# Impact of convection on the upper-tropospheric composition (water vapor/ozone) over a subtropical site (Réunion Island, 21.1°S-55.5°E) in the Indian Ocean

Damien Héron[1], Stéphanie Evan[1], Jérôme Brioude[1], Karen Rosenlof[2], Françoise Posny[1], Jean-Marc Metzger[3] and Jean-Pierre Cammas[1,3]

[1]LACy, Laboratoire de l'Atmosphère et des Cyclones, UMR8105 (CNRS, Université de La Réunion, Météo-France), Saint-Denis de la Réunion, France
[2]Chemical Sciences Division, Earth System Research Laboratory, NOAA, Boulder, CO, USA
[3]Observatoire des Sciences de l'Univers de La Réunion, UMS3365 (CNRS, Université de La Réunion, Météo-France), Saint-Denis de la Réunion, France

*Correspondence to*: Damien Héron (damien.heron@univ-reunion.fr)

**Abstract.** Observations of ozonesonde measurements of the NDACC/SHADOZ program and humidity profiles from the daily Météo-France radiosondes at Réunion Island (21.1°S, 55.5°E) from November 2013 to April 2016 were analyzed to identify the origin of wet upper tropospheric air masses with low ozone mixing ratio observed above the island, located in the South West Indian Ocean (SWIO). A seasonal variability in hydration events in the upper troposphere was found and linked to the convective activity within the SWIO basin. In the upper troposphere, ozone mixing ratios were lower (mean of 57 ppbv) in humid air masses (RH>50%) compared to the background mean ozone mixing ratio (73.8 ppbv). A convective signature was identified in the ozone profile dataset by studying the probability of occurrence of different ozone thresholds. It was found that ozone mixing ratios lower than 45 to 50 ppbv had a local maximum of occurrence between 10 and 13km in altitude, indicative of the mean level of convective outflow. Combining FLEXPART Lagrangian backtrajectories with METEOSAT 7 infrared brightness temperature products, we established the origin of convective influence on the upper troposphere above Réunion Island. It has been found that the upper troposphere above Réunion Island is impacted by convective outflows in austral summer. Most of the time, deep convection is not observed in the direct vicinity of the island, but more than a thousand of kilometers away from the island, in the tropics, either from tropical storms or the Inter Tropical Convection Zone (ITCZ). In November and December, the air masses above Réunion Island originate, on average, from Central Africa and the Mozambique Channel. During January, February the source region is the North-east of Mozambique and Madagascar. Those results improve our understanding of the impact of the ITCZ and tropical cyclones on the hydration of the upper troposphere in the subtropics in the SWIO.

# 1 Introduction

The variability of ozone in the tropical upper troposphere (10-16 km in altitude) is important for the climate as it influences the radiative budget (Lacis et al. 1990, Thuburn and Craig, 2002, Riese et al. 2012), and modifies the oxidizing capacity of the

atmosphere and the lifetime of other chemical species. The tropical ozone budget in the upper troposphere is influenced by stratospheric intrusions, convective transport from the surface, advection from mid-latitudes and chemical reactions.

Due to the complex interplay between dynamics and chemistry, tropical ozone concentrations observed by radiosondes at stations within the Southern Hemisphere ADditional OZonesondes (SHADOZ) network show large spatial and temporal variability (Thompson et al., 2003a; Fueglistaler et al., 2009). The average tropical ozone mixing ratio in the upper troposphere has a value of 40 ppbv, varying between 25 to 60 ppbv. Causes for the ozone variability in the tropics are particularly difficult to ascertain without a careful analysis of the processes involved in the observed variability (Fueglistaler et al., 2009). One
example is that the S shape found in the mean ozone profile in the SHADOZ stations over the Pacific, was interpreted by Folkins and Martin (2005) to be a consequence of the vertical profile of the cloud mass flux divergence.

In general, the impact of convection on the ozone budget in the tropical upper troposphere is not well established. Solomon et al. (2005) used a statistical method to characterize the impact of convection on the local ozone minimum in the upper troposphere above the SHADOZ sites within the maritime continent (Fiji, Samoa, Tahiti, and Java). They identified a minimum
of 20 ppbv of ozone in 40% of the ozone profiles. 20 ppbv corresponds also to the ozone mixing ratio in the local oceanic boundary layer. The sites are located in a convectively active region (Hartmann, 1994; Laing & Fritsch, 1997; Solomon et al., 2005; Tissier et al., 2016) and have a higher probability to be influenced by local deep convection than other SHADOZ sites.

Tropical convection can transport air masses from the marine boundary layer to the upper troposphere (Jorgensen and LeMone, 1989; Pfister et al., 2010) in less than a day. Because the ozone chemical lifetime is on the order of 50 days, air masses within
50 the convective outflow will retain the chemical signature of the boundary layer (Folkins et al., 2002, 2006).  Ozone can therefore be used as a convective tracer, and in doing such Solomon et al. (2005) estimated the mean level of convective outflow to be between 300 hPa and 100 hPa, or 8 and 14 km in altitude for SHADOZ stations located in the Western Pacific.

At present, little is known about the impact of convection on SHADOZ sites that are away from active convective regions. In the Southern Hemisphere, the position of Réunion Island (21.1°S-55.5°E) in the South-West Indian Ocean (SWIO, 10°S to
55 45°S and 40°E to 80°E) is particularly well suited to study the chemical composition of the troposphere over the Indian Ocean. During austral summer (November to April), the Inter Tropical Convergence Zone (ITCZ) moves closer to Réunion Island and convective activity in the SWIO is more pronounced with tropical cyclones forming in the region. Blamey and Reason (2012) estimated that the east of Mozambique Channel is the most convective zone of the region.

In this paper, we analyze ozonesonde measurements of the NDACC/SHADOZ program and humidity profiles from daily
Météo-France radiosondes from Réunion Island between November 2013 and April 2016 to identify the origin of wet upper tropospheric air masses with low ozone mixing ratio observed above the island and understand the role of transport, detrainment, and mixing processes on the composition of the tropical upper-troposphere over Réunion Island. We use infrared brightness temperature data from the METEOSAT 7 geostationary satellite to identify deep convective clouds over the SWIO region. The geographic origin of air masses measured by the radiosondes is estimated using Lagrangian backtrajectories
calculated by the FLEXible PARTicle (FLEXPART) Lagrangian Particle Dispersion Model (Stohl et al., 2005). Section 2 presents the radiosonde measurements, satellite products and FLEXPART model used in this study. Section 3 presents the seasonal variability in ozone/humidity as well as the convective influence on the radiosonde measurements. Results on the mean level of convective outflow and the convective origin of the air masses measured over Réunion Island are also presented in section 3. A summary and conclusions are given in section 4.

## 2 Measurements and Model

### 2.1 Ozone and water vapor soundings

The ozonesondes at Réunion Island are launched under the framework of the Network for the Detection of Atmospheric Composition Changes (NDACC) and the Southern Hemisphere ADditional OZonesondes (SHADOZ) programs. The SHADOZ project gathers ozonesonde and radiosonde (pressure, temperature, wind) data from tropical and subtropical stations (Sterling et al., 2018; Witte et al. 2017, 2018; Thompson et al., 2017). Between 2014 and 2016, 158 ozonesondes were launched at Réunion Island (almost 3 per month). The majority of the ozonesonde launches occur around 10 UTC at the airport (Gillot: 21.06°S, 55.48°E), located on the north side of the island. Balloons carry the ECC ozonesonde (Electrochemical Concentration Cell) in tandem with the Meteomodem M10 meteorological radiosonde. Smit et al. (2007) evaluate ECC-sonde precision to be better than ± (3-5) % and accuracy about ± (5-10) % below 30 km altitude.

In addition, we use data from operational daily meteorological Meteomodem M10 radiosondes launches performed by Météo-France (MF) at 12 UTC at the airport since 2013. The MF dataset provides relative humidity (RH) measurements with respect to water at a higher frequency than the SHADOZ data and this is important to study the day to day variability of the impact of convection on the upper troposphere. The Meteomodem M10 radiosondes provide measurements of temperature, pressure, RH with respect to water and zonal/meridional winds. We also calculated RH with respect to ice (RHi) when the temperature is below 0°C by using the formula of Hyland and Wexler (1983) for saturation vapor pressure over ice. We compared the M10 measurements with Cryogenic Frospoint Hygrometer (CFH) water vapor sondes when they are launched in tandem at the Maïdo Observatory (21.08°S, 55.38°E), located on the west coast of the island, 20 km away from the airport. Balloon-borne measurements of water vapor and temperature started in 2014 at the Maïdo Observatory on a campaign basis within the framework of the Global Climate Observing System (GCOS) Reference Upper-Air Network (GRUAN) network (Bodeker et al., 2015). The CFH was developed to provide highly accurate water vapor measurements in the Tropical Tropopause Layer (TTL) and stratosphere where the water vapor mixing ratios are extremely low (~2 ppmv). CFH mixing ratio measurement uncertainty ranges from 5% in the tropical lower troposphere to less than 10% in the stratosphere (Vömel et al., 2007; Vömel et al., 2016). Based on 17 (CFH+M10) soundings, we found that in the lower troposphere, below 5km, the mean RH difference is 1%. In the middle (5-10km) and upper troposphere (10-15km) the mean RH differences are 1.5 and 2.2% respectively. Near 15km in altitude, the M10 RH shows a dry bias with a peak difference of 3.7%.

For both MF and SHADOZ sondes, the average ascent speed of the balloon is 5m s$^{-1}$ and measurements are recorded every second so the mean native vertical resolution is around 5 m for both datasets. Vertical gaps as high as 500 m can occur in the two datasets and the native vertical resolution varies with altitude. Thus NDACC/SHADOZ ozonesonde and MF radiosonde data are interpolated to a regular vertical grid with a 200-m grid spacing.

As noted previously, this study focuses on austral summer conditions (November to April) and in particular the austral summer seasons 2013-2014, 2014-2015, and 2015-2016 (hereafter referred to as summer 2014, 2015 and 2016 respectively). Fig. 1 shows the NDACC/SHADOZ 2013-2016 seasonal average ozone mixing ratio profiles as well as the overall mean 4-year average. The 4-year average profile over 2013-2016 increases in the troposphere (from 25 ppbv at the surface to 200 ppbv at 17 km). In austral autumn (March, April and May) and winter (June, July and August) the ozone values are lower than the mean climatology in the troposphere above 3 km. Ozone values increase in the lower troposphere during the dry season (from May to September) when biomass burning plumes from southern Africa and Madagascar can be transported eastward and result in ozone production in the lower troposphere over the Indian Ocean (Sinha et al., 2004). The maximum of tropospheric ozone occurs in austral spring (September, October and November) at the end of the biomass burning season (which extends

from July to October, Marenco et al., 1990). Using ozonesonde and LIDAR from Réunion Island from 1998 to 2006, Clain et al. (2008) showed that the influence of stratosphere-troposphere exchange induced by the subtropical jet stream is maximum in austral winter (June to August) when the jet moves closer to the island. They established that the 4-10 km and 10-16 km altitude ranges can be directly influenced by biomass burning and stratosphere-troposphere exchange. The influence of stratosphere-troposphere exchange is in agreement with high ozone-low water vapor layers, which are ubiquitous over Réunion Island in austral winter. Austral summer (DJF) exhibits low ozone values, and in particular, below 3 km and between 9 and 14 km summertime ozone is at an annual low. The source of these low values will be discussed later in this paper.

## 2.2 METEOSAT 7 geostationary satellite data

METEOSAT 7 is a geostationary satellite positioned at the longitude 58°E that provides images for the Indian Ocean since December 2005. The Thermal Infrared channel (wavelength 10.5-12.5 µm) of the Meteosat Visible and Infra-Red Imager (MVIRI) instrument onboard METEOSAT 7, has a temporal resolution of 30 minutes and a horizontal resolution of 5 km at nadir. Here we use METEOSAT 7 hourly infrared brightness temperature product available from the ICARE data archive (ftp://ftp.icare.univ-lille1.fr).

We assume black body radiation (Slingo, 2004; Tissier et al., 2016) to estimate the brightness temperature. Young et al. (2013) have classified clouds in the tropics from the CloudSat and MODIS database for one year of observations (2007) over the 30°S-30°N latitude band. They established that cirriform clouds have, on average, higher brightness temperatures than deep convective clouds (respectively 268.5K against 228.5K, Fig. 5 of Young et al., 2013). Therefore, we identify deep convective clouds by selecting METEOSAT 7 pixels with brightness temperature lower than 230K. Minnis et al. (2008) used the Moderate Resolution Imaging Spectroradiometer (MODIS) 11µm IR channel data and data taken by the Cloud-Aerosol Lidar and Infrared Pathfinder Satellite Observations (CALIPSO) to investigate the difference between cloud-top altitude $Z_{top}$ and infrared effective radiating height $Z_{eff}$ for optically thick ice cloud (i.e. deep convective clouds). They found an error of 2km in the derived cloud top altitude from passive sensors for clouds higher than 14km in altitude, and an error of 1.25km below. This suggests that using a threshold of 230K to define deep convective clouds can induce an error in the selection of these clouds. Thin cirrus clouds could be included in our selection of deep clouds but it is difficult to say how much by using passive satellite sensors only. Additional measurements from active sensors such as CALIPSO would be required to distinguish between the deep convective cores inferred from passive infrared radiances and cold in situ formed cirrus clouds. However, this is beyond the scope of this study. We tested the sensitivity to the 230K threshold, and found that our definition of DCCO with a brightness temperature of 230K is a good compromise to distinguish deep convective clouds over land and ocean.

In order to fold FLEXPART weekly products with METEOSAT 7 infrared brightness temperature data, the latter dataset is interpolated to a regular latitude-longitude grid with a 1° resolution. In addition, for every day between November 2013 and April 2016, we create a map of the deepest convective clouds valid for the previous 7 days by accumulating their positions over the prior week. Thus, for each day we establish maps of Deep Convective Clouds Occurrence (DCCO) valid for the previous week as defined in equation 1.

$$DCCO[d,i,j] = \frac{1}{7 \times 24} \sum_{t=d-6}^{d} \sum_{h=0}^{23} N_t[h,i,j] \; with \; N_t[h,i,j] = \begin{cases} 1 \; if \; T_b < 230 \, K \\ 0 \; otherwise \end{cases} \quad (1)$$

In equation 1, DCCO is a function of day (d), latitude (i) and longitude (j). The term $N_t$ is the hourly highest cloud counter and Tb corresponds to METEOSAT 7 infrared brightness temperature. The weekly product is indicated by the sum between day "d" and day "d-6" (a total of seven days). We normalize DCCO by dividing the two sums in equation 1 by the total number of hourly METEOSAT 7 observations available during a week (i.e. 7x24 images). The mean DCCO map for the period of the study (summer seasons between November 2013 and April 2016) is shown on Fig. 2.

A weakness of the methodology relates to our treatment of convective tower anvils, which may have brightness temperatures colder than 230K. However, we assume that only convective centers correspond to cloud tops with a brightness temperature below 230K. We are using this assumption to identify the deep convective clouds and compare their distribution with the vertical transport from the boundary layer to the upper troposphere calculated by the FLEXPART model.

## 2.3 FLEXPART

To estimate the convective origin of mid to upper tropospheric air masses observed above Réunion Island, we use the FLEXPART Lagrangian particle dispersion model (Stohl et al. 2005). We use input meteorological fields from the ECMWF Integrated Forecast System (IFS, current ECMWF operational data) that have 137 vertical levels up to 0.01hPa. The vertical resolution varies from ~20 m near the surface, ~100 m in the low troposphere, ~300 m in the middle/upper troposphere and 500 m in the stratosphere. FLEXPART was driven by using operational ECMWF analysis at 00, 06, 12 & 18 UTC and the 3 & 9-hour forecast fields from the 00 and 12 UTC model analysis.

We use FLEXPART to calculate backtrajectories of particles from 3 bins at 1 km intervals in the upper troposphere (i.e. 10-11 km, 11-12 km,12-13 km) above Réunion Island. Vertical bins are defined between 10 and 13 km to trace the lower observed ozone values in the upper troposphere during austral summer (Fig. 1). The bins have a horizontal latitude-longitude resolution of 0.1°×0.1°. In every altitude bin, 10 000 trajectories are computed backward in time. Transport and dispersion in the atmosphere are done by the resolved winds and the subgrid turbulent parameterization. Two-week long backtrajectories are initialized every 3 hours (00, 03, 06, 09, 12, 15, 18 and 21 UTC) each day between 1 November 2013 and 31 December 2016. FLEXPART backtrajectories can then be processed in the form of a gridded output of the residence time. The residence time field was reported on a regular 0.5°x0.5° output grid every 3 hours. The resolution of the gridded output is independent from those of the meteorological input. Therefore, we use 0.25°x0.25° operational ECMWF input fields to compute the backward trajectories and the resulting residence time is reported on a regular 0.5°x0.5° output grid. The residence times of particles indicate where and for how long air masses sampled over the observation site have resided in a given atmospheric region (lower troposphere, planetary boundary layer, …) along the backtrajectories (Stohl et al., 2005). The residence time of the backtrajectories are computed on 1°x1° grid cells using the FLEXPART model output values and summed over 24 hours to provide a daily estimate of the source regions. We define the daily fraction of residence time in the lower troposphere (RTLT, equation 2), as the residence time of air masses that were in the troposphere below 5 km divided by the total residence time in the troposphere. RTLT is a function of day (d), latitude (i) and longitude (j). The convective origin of an air mass observed in the upper troposphere can be inferred by high values of RTLT, i.e. the air mass was in the lower troposphere below 5 km for a significant amount of time compared to the total residence time spent in the whole troposphere. The threshold at 5 km to define the lower troposphere was chosen to take into account the convective transport of air masses from the boundary layer and subsequent in-cloud mixing during the ascent from the lower troposphere to the upper troposphere. The location, intensity and vertical extent of deep convection in the FLEXPART model is determined by the calculation of a CAPE and the atmospheric thermodynamic profile using the meteorological fields from ECMWF. The trajectories are then redistributed

vertically by a displacement matrix. Hence, the accuracy of the convective cells' location will be driven by the convective cells' locations within the ECMWF model output.

$$RTLT[d,i,j] = \frac{1}{T_{total}} \sum_{z=0}^{5km} \sum_{h=1}^{8} T_d[h,i,j,z] \quad with$$

$$T_d[h,i,j,z] = residence\ time\ in\ (i,j,z), for\ back\ trajectories\ initialized\ the\ "d"\ day\ at\ "h"\ hours$$

$$T_{total}[d] = \sum_{i,j,z} \sum_{h=1}^{8} T_d[h,i,j,z] = the\ total\ residence\ time\ of\ trajectory\ initialized\ the\ "d"\ day \qquad (2)$$

## 3 Results

### 3.1 Seasonal variability of relative humidity

Fig. 3 shows the time series of vertical profiles of RH from 2014 to 2016. High RH values (above 80%) observed below 2 km are typical values of the tropical humid marine boundary layer (Folkins and Martins, 2004). The mean value of RH in the upper troposphere (10-13 km) ranges from ~10% during the dry season (austral winter, May to October) to 40% during the wet season (austral summer, November to April) throughout the year. The troposphere between 2 and 10 km shows higher values of RH (mean of 37%) during austral summer than during the austral winter (mean of 15%). Above the 0°C isotherm,

the RHi contour (RHi>100%) shows a low proportion of potential cirrus clouds in the most hydrated profiles.

Higher values of RH ~60% from the boundary layer to the upper troposphere can be observed sporadically during austral summer. These higher values of RH are related to convective events (e.g. tropical thunderstorms and/or cyclones) in the vicinity of the island. Other high values of RH (>40%) in the upper troposphere also appear and they do not seem directly connected to local convection over Réunion Island. We define these higher values of RH in the upper troposphere as "upper-tropospheric

hydration events". We will later show that these upper-tropospheric hydration events are associated with convective detrainment of air masses in the upper troposphere and their subsequent long-range transport to Réunion Island. Profiles with a RHi > 100% are represented by black isocontours on Fig. 3. RHi values ≥ 100% could be an indication of the presence of cirrus clouds. However additional remote-sensing instruments (e.g. Lidar/Radar) would be needed in addition to the radiosonde measurements of RH to truly assess the presence of these clouds. Fig. 4 shows the daily evolution of mean upper-tropospheric

(10-13km) RH from September 2013 to July 2016. The RH values are color-coded according to the RHi (top) and the water vapor mixing ratio (bottom) values. The different RHi/water vapor mixing ratio ranges used on Fig. 4 correspond to our definition of dry profiles (in blue), wet profiles (orange) and supersaturated profiles (red). To distinguish the effect of temperature and water vapor on RH/RHi values, we computed the water vapor mixing ratio (WV) for each profile between September 2013 to July 2016. We found a mean 10-13km WV of 121ppmv over this period. The value is in agreement with a

climatological WV value computed with Microwave Limb Sounder (MLS) v4.2 water vapor data for 2005-2017. We calculated a MLS climatological WV profile for a region of 5°x5° surrounding Réunion Island. The MLS climatological WV profile at 261 hPa (~10km) is 116 ppmv and agrees with the mean upper-tropospheric value of 121 ppmv inferred from the radiosonde data. The top panel on Fig. 2 shows that 91% of profiles with RH>50% are associated with high RHi > 80%. These events have also a high WV and indicate a hydration rather than a cooling effect on the high RH/RHi values. Some hydrated profiles

(RH > 50%, 9% of the profiles) with a low RHi (< 80%) are present in January 2016 and this could be linked to a cooling of the upper troposphere. 27% of the hydrated profiles (RH>50%) correspond to supersaturated RHi (RHi>100%) and occur mostly in 2015 and 2016. The RH of averaged water vapor mixing ratio (WV, 121 ppmv) is compared to the most hydrated profile (WV>302.5 ppmv). There are few events with WV>121ppmv in winter, the averaged water vapor in winter is around 66 ppmv against 182 ppmv in summer. Peak values of RH as high as ~60% are observed and linked to a net hydration of the upper troposphere (WV>304ppmv). Their occurrence varies from 2014 to 2016. As the distinction between high RHi (>80%) and low RHi (<80%) is similar to the distinction between hydrated profiles (RH>50%) and dry profiles (RH<50%), RH is used subsequently, instead of RHi, to study convective effects on the hydration of the upper troposphere above Réunion Island.

It is known that the El-Niño–Southern Oscillation (ENSO) can affect convective activity over the SWIO (e.g. Ho et al., 2006; Bessafi and Wheeler, 2006). The NOAA Climate Prediction Center Ocean Niño index (ONI, http://origin.cpc.ncep.noaa.gov/products/analysis_monitoring/ensostuff/ONI_v5.php), which is based on SST anomalies in the Niño 3.4 region, was equal to -0.4 in austral summer 2014 (ENSO neutral conditions), +0.6 in austral summer 2015 (weak El Niño) and +2.2 in austral summer 2016 (strong El Niño). With an increase in sea surface temperatures (SST) during El Niño events, convective activity over the SWIO is enhanced (Klein et al. 1999). At the same time, El Niño events can increase the vertical wind shear over the SWIO, which could reduce the intensification of tropical cyclones (Ho et al., 2006) and so increases the number of storms that do not reach the tropical cyclone stage (in the SWIO a storm is classified as a tropical cyclone when 10-minute sustained winds exceed 118 km.h$^{-1}$).

The differences in humidification in the upper troposphere can also be affected by the Madden Julian Oscillation (MJO). To define the state of the MJO, we used the Real-time Multivariate MJO (RMM) indices RMM1 and RMM2 data from the Australian Bureau of Meteorology (http://www.bom.gov.au/climate/mjo/graphics/rmm.74toRealtime.txt). RMM1 and RMM2 are based on a combined empirical orthogonal function analysis of 15°S to 15°N averaged Outgoing Longwave Radiation in addition to zonal winds at 850 and 200 hPa (Wheeler and Hendon, 2004). The MJO cycle, as defined by RMM1 and RMM2, can be split up into eight phases with phases 2 and 3 corresponding to a MJO convective center over the Indian Ocean. The square root of the square summation of RMM1 and RMM2 represents the MJO amplitude. The MJO is defined as active when its amplitude is greater than 1.

During the 3 austral summers studied, the MJO was active over the Indian Ocean for a similar number of days (14%, 18%, 18% of the time in austral summers 2014, 2015 and 2016 respectively). The averaged upper tropospheric RH for an active MJO over the Indian Ocean is 30%, almost the same as the climatological RH over the period November 2013 to April 2016 (cf. Fig. 5). During some of these MJO events there was an increase in RH, e.g. 5-11 December 2013 (50%), 3-5 November 2015 (46%), 13-20 January 2016 (52.4%) and 1-3 February 2016 (54.8%). Garot et al. (2017) studied the evolution of the distribution of upper-tropospheric humidity (UTH) over the Indian Ocean with regard to the phase of the MJO (active or suppressed). They used RH (with respect to water) measurements from the Sounder for Atmospheric Profiling of Humidity in the Intertropics by Radiometry (SAPHIR)/Megha-Tropique*s* radiometer, RH measured by upper-air soundings, dynamic and thermodynamic fields produced by the ERA-Interim model and the cloud classifications defined from a series of geostationary imagers to assess changes in the distribution of UTH when the development of MJO takes place in the Indian Ocean. There is a strong difference in the distribution of UTH according to the phase of MJO (active or suppressed). During active (suppressed) phases, the distribution of UTH measured by SAPHIR was moister (drier). However, their study focused on the equatorial (8°S-8°N) Indian Ocean region whereas we are investigating upper-tropospheric RH distribution over a subtropical site. The MJO is the main driver of the fluctuations of tropical weather on weekly to monthly time scales over the Indian Ocean. Thus, it can influence convective activity (e.g. tropical cyclones) over the basin and the subsequent cloudiness and upper tropospheric RH (via transport of moisture). A clearer explanation on the interplay between ENSO/MJO and upper tropospheric humidity

over a subtropical site such as Réunion Island would require the analysis of additional years, but this is out of the scope of this study.

Austral summer 2014 (Fig. 3 & 4) is affected by three tropical cyclone events. Overall, summer 2014 is the driest of the three years. Consistent with a higher ONI at +0.6, higher convective activity is observed in austral summer 2015. The outflow from two tropical cyclones affected Réunion Island: Bansi from January 9 to 19 and Chedza January 13 to 22. Higher convective activity was also observed in February and March 2015. In 2016, associated with a strong El Niño event (ONI=+2.2), there is an increase in convective activity as compared to austral summers 2014 and 2015. Fig. 4 shows that austral summer 2016 is associated with higher RH in the upper troposphere (Fig. 4). Previous studies have shown a correlation between intense El Niño events and an increase in ITCZ precipitation over the SWIO (Yoo et al., 2006, Ho et al. 2006). We will later show that the majority of the austral summer 2016 upper tropospheric hydration events are associated with the convective activity located in the ITCZ.

Fig. 5 shows the histogram of RH between 10 and 13 km for the three austral summer periods (2014, 2015 and 2016). We choose a RH value of 25% (corresponding to the median of the distribution) to characterize the upper tropospheric background, which should be dry without the effect of convective hydration. In the rest of the study, a RH threshold of 50% is used to isolate upper tropospheric air masses that have likely been affected by deep convection. A threshold on RH had to be chosen to isolate the RH/ozone profiles that were most likely impacted by convection. The average water vapor mixing ratio between 10 and 13 km in austral summer (182ppmv) is larger than in austral winter (65ppmv), probably due to the effect of deep convection and associated moisture transport and cloudiness. The average RH of air masses with water vapor mixing ratios greater than 182ppmv is 48.8%. Thus, a RH threshold of 50% is used to isolate upper tropospheric air masses that may have been affected by deep convection in the rest of the study.

## 3.2 Convective influence on the upper troposphere

In this part of the study, we use the NDACC/SHADOZ dataset to analyze the convective influence on air masses observed above Réunion Island. The 2013-2016 NDACC/SHADOZ ozone dataset has a mean background value of 81 ppbv in the upper troposphere (average ozone mixing ratio between 10 and 13 km). Fig. 6 shows the ozone distributions for the lower troposphere (below 5km, green bars on Fig. 6) and the upper troposphere (10-13 km, grey bars on Fig. 6). 76.8% of the lower tropospheric ozone data have values ranging from 15 to 40 ppbv. These values agree with ozone mixing ratios typically observed for air masses in the marine boundary layer (20 ppbv), we note that the values larger than 20 ppbv can be explained by mixing with air masses of the tropical free troposphere with climatological higher ozone content. In the upper troposphere, ozone mixing ratios range from 30 to 110 ppbv (Fig. 6). To estimate the average residence time in the upper troposphere, we analysed the evolution of RTLT for different backtrajectory durations (not shown). RTLT from 46-hour backtrajectories is mostly located in the vicinity of Réunion Island, and the North-East of Madagascar. The 96-hour RTLT pattern is significantly different and spreads over the Eastern and Northern regions of Madagascar for 2015 and 2016, and also West of Madagascar in 2014. The pattern of 120-hour and 168-hour RTLT is roughly similar to the 96-hour RTLT, except that RTLT is more spread over the North-East and West of Madagascar. It means that most of the humid air masses reaching the 10-13km layer above Réunion Island were embedded in convective clouds and were transported from the lower troposphere to the upper troposphere within 96 hours. The spread in the RTLT product from 96 hours to 168 hours backward in time is the result of horizontal atmospheric transport in the lower troposphere. Therefore, we can estimate an average time of transport between the main convective sources and the upper troposphere over Réunion Island to be 96 hours.

For the upper troposphere, we further consider the ozone distribution for humid air masses by using a RH threshold of 50%. We performed sensitivity tests by using RH thresholds ranging from 40 to 55% and found that the ozone distribution in the upper troposphere is very similar for these different RH thresholds (not shown). One main mode appears in the ozone distribution for air masses with RH > 50% (blue bars on Fig. 6) that is centered around 45 ppbv (56.4% of data are between 30 and 57.5 ppbv) As explained previously, the mode centered around 45 ppbv in the wet distribution may be associated with vertical transport of low-ozone air masses from the marine boundary layer to the upper troposphere and subsequent mixing with tropospheric air masses with higher ozone content along their pathway.

Ozone mixing ratios higher than 70 ppbv are observed less frequently in the moist upper troposphere (16% of the observations) than in the total distribution (43% of the observations). However, the average ozone mixing ratio in the humid upper troposphere is on average higher than the ozone mixing ratio observed in the lower troposphere (45 ppbv against 31.7 ppbv respectively). This again agrees with a convective transport pathway from the marine boundary layer to the upper troposphere and mixing along the pathway.

As suggested in Fig. 2, and later discussed in section 3.5, the deep convection that may commonly influence the upper troposphere above Réunion Island is not directly in the vicinity of the island but further north in the ITCZ region. The difference in the ozone signature in the low-troposphere (31 ppbv) and directly above Réunion island (45 ppbv) suggests that mixing processes occurred during the long-range transport through the upper troposphere enriched in ozone (~81 ppbv) between the convective region and Réunion Island. Another explanation could be that land-based convection (from Madagascar or Africa) lifted air masses enriched in ozone from the boundary layer.

### 3.3 Level of convective outflow

Solomon et al. (2005) have studied ozone profiles at several tropical sites in the Southern Hemisphere to characterize the impact of deep convection on the ozone distribution in the tropical troposphere. They studied 6 years of measurements (1998 to 2004) from different stations of the SHADOZ network. In the Solomon et al. study, 40% of the ozone profiles over the Western Tropical Pacific (WTP) stations (Fiji, Samoa, Tahiti, and Java) have ozone mixing ratios lower than 20 ppbv within the upper troposphere (10 to 13 km). 20 ppbv is the average ozone mixing ratio found in the clean marine boundary layer of the WTP. The WTP is the most active convective basin of the Southern Hemisphere due to warmer SSTs in this region (Hartmann, 1994; Laing and Fritsch, 1997; Solomon et al., 2005; Tissier et al., 2016). Hence, ozone profiles in the WTP have a higher probability of being influenced by recent and nearby convection than other SHADOZ stations. This explains the weaker probability ozone mixing ratios lower than 20 ppbv in the upper troposphere for other stations which are located further from the ITCZ region.

Fig. 7 shows fractions of the ozone distribution lower than different ozone mixing ratios (25, 40, 45, 50, 55 and 60 ppbv) for ozone profiles observed during the austral summer seasons of 2013 to 2016. A low probability of measuring ozone mixing ratios lower than 25 ppbv is found at the top of the boundary layer. Furthermore, none of the ozone profiles have a mixing ratio lower than 20 ppbv between 8 and 13 km, confirming the results of Solomon et al. (2005), and less than 12% have an ozone mixing ratio lower than 40 ppbv. However, the fraction of ozone profiles displays a maximum of occurrence for ozone thresholds at 45 (22%), 50 (27%) and 55 ppbv (35%) between 10 and 13 km, corresponding to the altitude of the mean level of convective outflow found in Solomon et al. (2005).

We will show in the subsequent sections that the ozone chemical signature of convective outflow diagnosed from Fig. 7 is mainly associated with air masses detrained from the ITCZ. In comparison to the WTP region, the ITCZ is primarily located

north of Réunion Island (Schneider 2014), even in austral summer (Fig. 2). Considering that Réunion Island is farther from the ITCZ than the stations in the WTP, a longer time for long-range transport to occur is needed from the convective region to Réunion Island and thus mixing between low-ozone air masses in the boundary layer with high-ozone air masses in the upper troposphere can explain the values observed in the upper-troposphere over Réunion Island. Moreover, photochemical production of ozone during long-range transport after convective entrainment can increase the ozone of an air mass (Wang et al., 1998).

### 3.4 Origin of convective outflow observed above Réunion Island

### 3.4.1 Tropical Cyclone Hellen as a case-study

Hellen was a tropical cyclone that formed in the Mozambique Channel and was named on 26 March 2014. It became a category 4 tropical cyclone on the Saffir-Simpson scale on 30 March 2014. After reaching its maximum stage on 30 March 2014, Hellen was classified as category 1 on the Saffir-Simpson scale and was 1200 km away from Réunion Island at the time of the sounding on 31 March 2014 at 12UTC. Although tropical cyclone Hellen is not the most influential cyclone on the upper troposphere above Réunion Island, it is a relevant case-study as it is representative of tropical cyclones that form in the Mozambique Channel for the SWIO region. In addition, this system had a clear signature in the RH profile in the upper troposphere (relative maximum of RH of 60% at 11 km altitude on Fig. 8c, red curve). Since RHi is around 100% at 11 km and the decrease in humidity below the layer is slower than above the layer, it probably indicates a hydration effect due to sedimented ice crystals.

Patterns of the RTLT (fraction of residence times in the lower troposphere, see section 2.3 for definition) during the week before 31 March 2014 for air masses sampled in the upper troposphere above Réunion Island are displayed on Fig. 8a. RTLT can be considered as a map of density probability function of origin of the thousands of trajectory particle source locations in the lower troposphere. High values of RTLT (filled contours on Fig. 8a) are observed over the Mozambique Channel and are coincident with the best track of tropical cyclone Hellen (red curve on Fig. 8a and 8b). Thus, the FLEXPART backward trajectories indicate that the air mass sampled on 31 March 2014 above Réunion Island spent a significant amount of time in the lower troposphere during the previous week while tropical cyclone Hellen was intensifying over the Mozambique Channel. Additionally, the high values of RTLT coincide with a high weekly mean convective cloud cover (DCCO see section 2.2 for definition) for the same week (Fig. 8b). The weekly DCCO was higher over the Mozambique Channel in agreement with the presence of the tropical cyclone in this region during the week preceding 31 March 2014.

The two maps of RTLT and DCCO roughly display the same pattern (maximum above the Mozambique Channel). A detailed analysis of RTLT was performed for tropical cyclone Hellen with different residence time in the lower troposphere with 48 hours, 96 hours, 120 hours, and 168 hours FLEXPART backtrajectories (not shown). After 48 hours, no contribution in RTLT is found. After 96 hours, the RTLT is located north of the storm track, within the convective region of tropical cyclone Hellen. After 120 and 168 hours, an anti-clockwise dispersion toward Africa, outside the convective cells is found. It represents the fraction of air masses in the lower troposphere that was advected toward the convective clouds before reaching the 10-13km altitude range. Hence, the collocation of RTLT with DCCO depends on the collocation of the convective regions in FLEXPART+ECMWF and METEOSAT7, but also on the duration of the backtrajectories.

By combining the two products of FLEXPART derived RTLT and METEOSAT 7 DCCO, we can thus infer that the air mass sampled in the upper troposphere over Réunion Island on 31 March 2014 was in the lower troposphere over the Mozambique Channel the week before. This air mass was located 1500 km away from Réunion Island and was transported from the lower

troposphere to the upper troposphere by deep convective clouds within tropical cyclone Hellen and then advected eastward toward Réunion Island. Hence this specific case study illustrates the ability of the FLEXPART model to track the convective origin of air masses in the upper troposphere above Réunion Island.

### 3.4.2 Impact of convection on RH variability.

In this section, we will identify which tropical cyclones have influenced the upper troposphere above Réunion Island. We display on Fig. 9 the trajectories of 23 tropical cyclones (8 in 2014, 9 in 2015 and 6 in 2016) that were within a 2100 km radius around Réunion Island, representing 74% of tropical cyclones that developed within the SWIO basin between summer 2014 and 2016 (from November 2013 to April 2016). Outside the 2100 km radius, the influence of tropical cyclones (TC) on Réunion Island's upper troposphere is found to be limited (not shown).

There is significant variability in the number of SWIO cyclones that traverse (or maybe form in) the Mozambique Channel in a given year. For 2014 there were 3 (out of 8 for the SWIO), in 2015 2 (out of 9) and for 2016, none (out of 6). Near Réunion Island, a similar activity is found during the three summer seasons (about 2 cyclones per year in the direct vicinity of the island). In 2014, tropical cyclone Bejisa was the only cyclone that directly impacted Réunion Island. During the three austral summer seasons of 2014, 2015 & 2016, half of the tropical cyclones formed Northeast of Réunion Island (12 in total).

In order to determine the tropospheric origin of upper tropospheric air masses observed over Réunion Island during summer 2014, 2015 & 2016 (Fig. 3), we integrated the RTLT gridded over the domain of study (1° latitude-longitude resolution) to define the spatially integrated quantity sRTLT (Fig. 10). We calculated a similar product for the middle troposphere (sRTMT, 5-10km). A peak in the time series of sRTLT in Fig. 10 means that an event, associated with a deep vertical transport from the lower troposphere to the 10-13km altitude range, has increased the lower tropospheric origin of air masses measured in the upper troposphere above Réunion island. Hence, we integrated the values of the RTLT folded with DCCO (RTLTxDCCO) to obtain the probability of convective origin of each air mass (Fig. 10). When and where this cumulative probability is not null, the product RTLTxDCCO points at the convective events that most likely hydrated the upper troposphere over Réunion Island. If a peak in sRTLT is correlated with a peak in RTLTxDCCO, it means that the lower-tropospheric origin estimated by FLEXPART simulations corresponds to convective clouds observed by the METEOSAT 7 satellite. Finally, sRTLT and RTLTxDCCO are compared to the average upper-tropospheric (10-13km) water vapor mixing ratio over Réunion Island (Fig. 10). The RTLTxDCCO allows to identify the tropical cyclones that have hydrated the upper troposphere over Réunion Island, i.e. TCs Bejisa, Deliwe, Guito and Hellen in 2014 (B, D, G and H: upper panel of Fig. 10); Bansi, Chedza, Fundi and Haliba in 2015 (B, C, F and H: middle panel of Fig. 10); Corentin and Daya in 2016 (C, D: lower panel of Fig. 10). Note that the absence of soundings between 26 and 31 December 2015 prevents diagnosing the influence of tropical storm A2 in 2015. Among these cyclones, 3 of them (Bejisa in 2014; Bansi, and Haliba in 2015) had trajectories close to Réunion Island, 6 of them (Deliwe, Guito, Hellen in 2014; Chedza and Fundi in 2015 and Daya in 2016) had trajectories west of Réunion Island, and none had a trajectory east of Réunion Island. The results are consistent with the study of Ray and Rosenlof (2007). Using Atmospheric Infrared Sounder (AIRS) and MLS satellite data, Ray and Rosenlof (2007) estimated the enhancement of water vapor due to 32 typhoons (Western Pacific) and 9 hurricanes (Northern Atlantic) at 223 hPa (~11km). They found an enhancement of up to 60 to 70 ppmv, within a 500 km radius north of the tropical storm centers, where the highest water vapor enhancement was found. The convective outflow of tropical cyclones that impacted the upper troposphere over Réunion Island was located south of the cyclone centers, the most hydrated part of the tropical cyclones in the Southern Hemisphere according to Ray and Rosenlof (2007).

sRTMT in Fig. 10 represents the origin in the middle troposphere. An increase in sRTMT is associated with a vertical transport in the troposphere weaker than events that increase sRTLT, such as deep convection. A study of Schumacher et al. (2015) has shown that vertical transport within stratiform clouds can reach 10 m s$^{-1}$ below 7 km and has a slower ascent rate (<0.5 m s$^{-1}$) up to 10km. It suggests that the variability in sRTMT signature may be related to differences in the impact of stratiform clouds on water vapor mixing ratio in the upper troposphere (10-13km). Fig. 10 shows a higher correlation between water vapor mixing ratio variability in the upper troposphere and the sum of sRTLT and sRTMT than sRTLT or sRTMT taken individually. sRTLT+sRTMT and the upper tropospheric WV have a squared linear correlation coefficient of 0.46, while RTLT or RTMT have a squared linear coefficient of 0.23 and 0.42 respectively with upper tropospheric WV mixing ratio. It indicates that water vapor transport occurred both from the lower troposphere (e.g. by deep convection) or from the middle troposphere (e.g. by large-scale uplift of air masses associated with stratiform clouds) toward the upper troposphere. While the relative contribution of RTLT and RTMT varies over the summer seasons, the highest peaks in WV mixing ratio are associated with peaks in RTLT, due to convective transport associated with the passage of tropical cyclones.

### 3.4.3 Geographic origin of the convective outflow

Fig. 11 shows the monthly averaged maps of the product between DCCO and RTLT, which represents the probability of convective influence from a given region on the upper troposphere above Réunion Island. At the beginning of the austral summer seasons (November 2013, 2014 and 2015), the main convective regions that influence the upper troposphere above Réunion island are located in central Africa (Congo and Angola). Then from November to January, the influential convective region moves to the east towards the Mozambique Channel. For seasons 2014 and 2015, most of the influential convective regions are linked to cyclonic activity. TC Bejisa (B-2014, near Réunion Island) and TC Deliwe (D-2014, in the Mozambique Channel) were the most influential convective events in January 2014. For January 2015, two tropical cyclones were active in the SWIO, Bansi (B-2015, near Réunion Island) and Chedza (C-2015, Mozambique Channel). In February 2014, three cyclones formed in the SWIO basin. Despite the short distance from the island, TC Edilson (E-2014) did not have a significant influence on the upper troposphere above Réunion Island, while more remote tropical cyclone such as Guito (G-2014, in Mozambique Channel) significantly hydrated the upper troposphere above Réunion Island. In March 2014, the little patch in the Mozambique Channel was directly linked with the TC Hellen (H-2014). In February 2015 TC activity decreased and convection over Madagascar hydrated the upper troposphere above Réunion Island. TC Haliba (H-2015) caused the maximum value of DCCOxRTLT observed east of Madagascar in March 2015 (red contour on Fig. 11 for March 2015). There were fewer tropical cyclones (Fig. 10 & 11) that influenced Réunion Island in 2016, but there was nonetheless intense convective activity over the SWIO. In austral summer 2016, convective activity was more spread across the SWIO and Southern Africa.

### 4 Summary and conclusion

We analyzed ozonesonde measurements from the NDACC/SHADOZ program and humidity profiles from the daily Météeo-France radiosondes at Réunion Island between November 2013 and April 2016 to identify the origin of wet upper tropospheric air masses with low ozone mixing ratio observed above the island, located in the subtropics of the SWIO basin.

A seasonal variability in hydration events in the upper troposphere was found. The variability was linked to the seasonal variability of convective activity within the SWIO basin. An increase in the convective activity in austral summer 2016 (a strong El Niño year) compared to austral summers 2014 and 2015 was associated with higher upper tropospheric hydration. In the upper troposphere, ozone mixing ratios were lower (mean of 57 ppbv) in humid air masses (RH>50%) compared to the background mean ozone mixing ratio (73.8 ppbv).

A convective signature was identified in the ozone profile dataset by studying the probability of occurrence of different ozone thresholds. It was found that ozone mixing ratios lower than 45 to 50 ppbv had a local maximum of occurrence near the surface and between 10 and 13km in altitude, indicative of the mean level of convective outflow, in agreement with Solomon et al. (2005) and Avery et al. (2010).

Combining FLEXPART Lagrangian backtrajectories with METEOSAT 7 infrared brightness temperature products, we established the origin of convective influence on the upper troposphere above Réunion Island. We found that the ozone chemical signature of convective outflow above Réunion Island is associated with air masses detrained from the ITCZ located northwest of the island and tropical cyclones in the vicinity of the island (2100 km around the island). A higher correlation between tropical cyclone activity and high upper tropospheric RH values was found in austral summers 2014 and 2015. It was
found that isolated convection within the ITCZ was more pronounced in 2016 (most likely due to the strong El Nino) and as a result the vertical transport associated with these isolated convective clouds were misrepresented in the 0.25x0.25° meteorological fields used to drive the FLEXPART model. For austral summers 2014 and 2015, the FLEXPART model is able to trace back the origin of upper tropospheric air masses with low ozone/high RH signatures to convection over the Mozambique Channel/Madagascar and within tropical cyclones.

Hence, it has been found that the upper troposphere above Réunion Island is impacted by convective outflows in austral summer. Most of the time, deep convection is not observed in the direct vicinity of the island, as opposed to the Western Pacific sites in Solomon et al. (2005) study, but more than a thousand kilometers away from the island in the tropics either from tropical storms or the ITCZ. In November and December, the air masses above Réunion Island originate, on average, from Central Africa and the Mozambique Channel. During January, February the source region is the Northeast region of
Madagascar and the Mozambique Channel.

The average chemical ozone signature of convective outflow was found to be 45 ppbv between 10 and 13km in altitude, which differs from the 20 ppbv threshold used in Solomon et al. (2005). The higher threshold can be explained by vertical transport of low-ozone air masses from the marine boundary layer to the upper troposphere and subsequent mixing with tropospheric air masses with higher ozone content along their pathway when advected over more than a thousand kilometers.

**Data availability**

METEOSAT 7 data used in this study are available at http://www.icare.univ-lille1.fr/archive. The NDACC/SHADOZ ozone measurements for Réunion Island are available at https://tropo.gsfc.nasa.gov/shadoz/Reunion.html. The FLEXPART Lagrangian trajectories can be requested from Stephanie Evan (stephanie.evan@univ-reunion.fr).

**Author contributions**

All authors contributed to the paper. DH wrote the manuscript with contributions from SE, JB, KR, JPC. JMM and FP performed the ozone radiosonde measurements. SE and JB performed the FLEXPART simulations. DE processed the radiosonde and FLEXPART data. All authors revised the manuscript draft.

**Competing interests**

The authors declare that they have no conflict of interest.

**Acknowledgments**

 OPAR (Observatoire de Physique de l'Atmosphère à La Réunion, including Maïdo Observatory) is part of OSU-R (Observatoire des Sciences de l'Univers à La Réunion) which is being funded by Université de la Réunion, CNRS-INSU, 720 Météo-France, and the french research infrastructure ACTRIS-France (Aerosols, Clouds and Trace gases Research Infrastructure). This work was supported by the French LEFE CNRS-INSU Program (VAPEURDO).

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

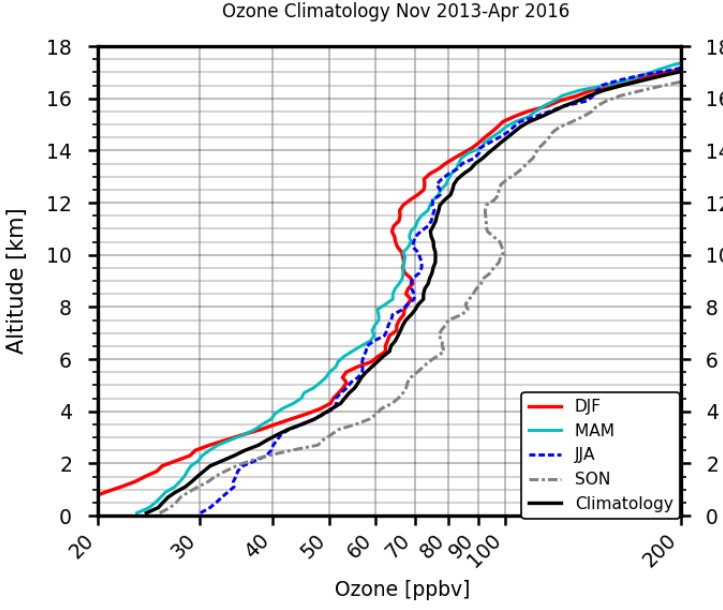

**Figure 1: Climatological mean ozone profile (from November 2013 to April 2016) in black. 2013-2016 seasonal mean ozone profile for summer (December, January, February) in red, spring (March, April, May) in cyan, winter (June, July, and august) in blue and autumn (September, October, November) in grey.**

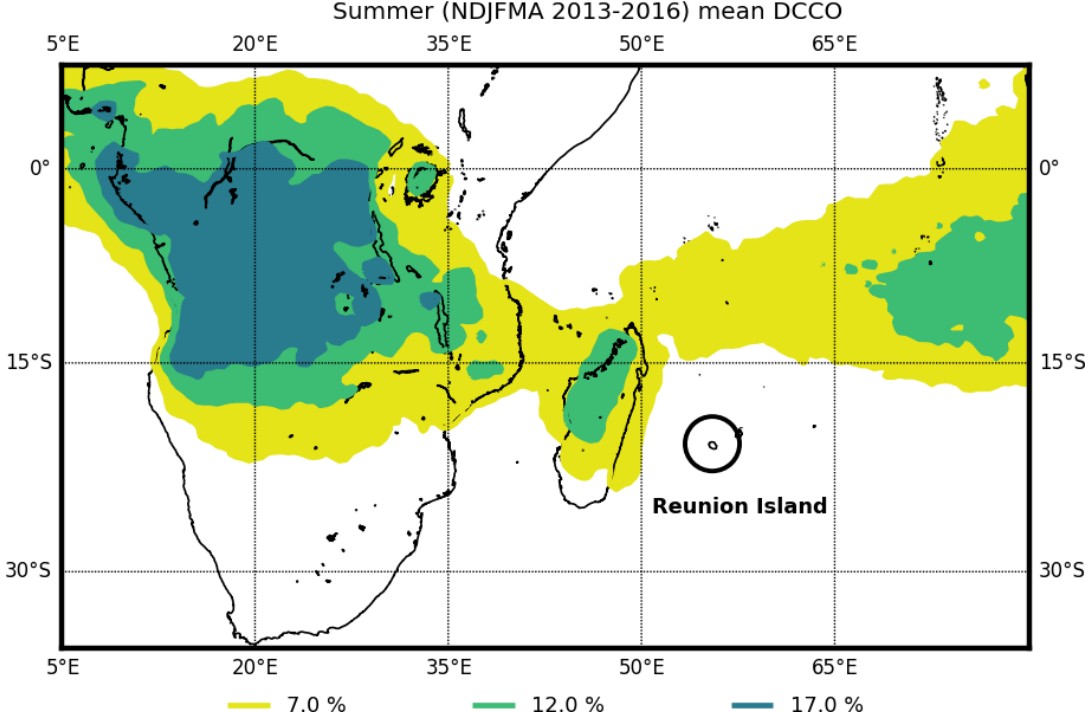

**Figure 2: Average map of the deepest convective cloud occurrence (DCCO) for austral summer conditions (NDJFMA from November 2013 to April 2016). The yellow contour is for DCCO > 7%, the green contour for DCCO > 12% and the dark green contour is for DCCO > 17%.**

RH evolution

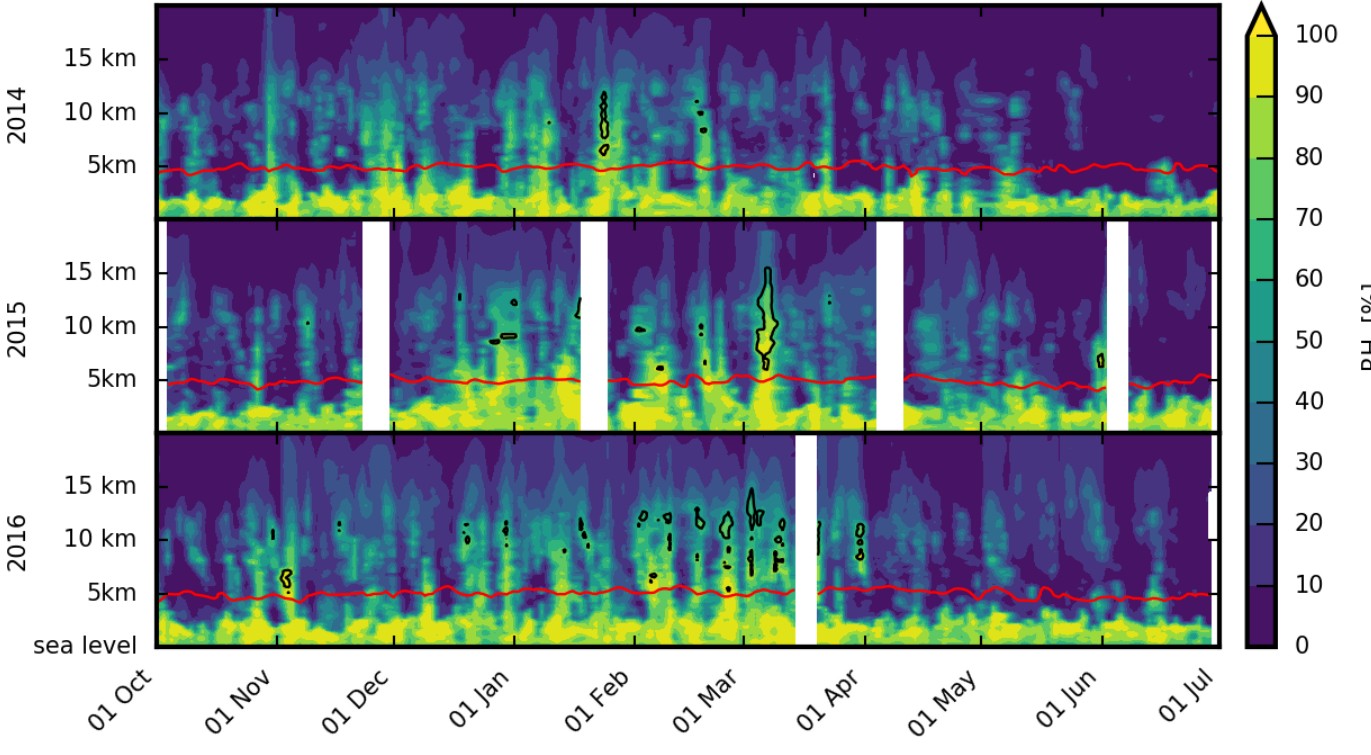

605

**Figure 3: Time height cross-sections of day-to-day variations of Relative Humidity (RH) for Oct 2013-July 2014 (upper panel), Oct 2014-July 2015 (middle panel) and Oct 2015-July 2016 (bottom panel). The RH field is interpolated in time except when 5 days of data are missing. Black contours are for RHi>100%, and the red line corresponds to the 0°C isotherm.**

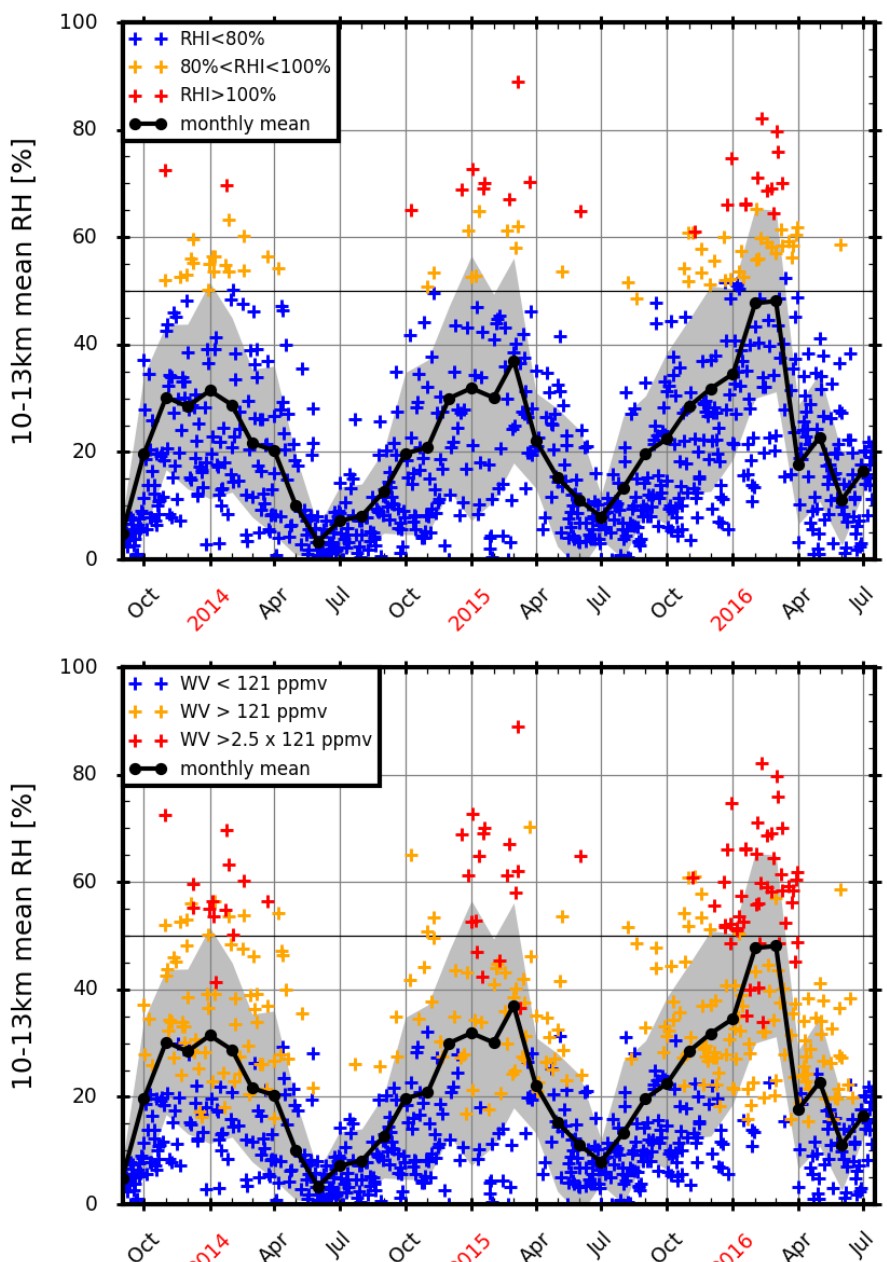

Figure 4: Top, daily evolution of mean upper tropospheric (10 to 13 km) RH for the period September 2013 to July 2016. The RH (with respect to water) values are color coded according to the values of RHi. In blue RH values for RHi <80%, in orange RH values for 80% <RHi < 100%, and in red RH values for RHi> 100%. Bottom, daily evolution of mean upper tropospheric (10 to 13 km) RH for the period September 2013 to July 2016. The RH (with respect to water) values are color coded according to the values of water vapor mixing ratios (WV). The mean 10-13km WV is 121 ppmv. In blue RH values for WV < 121 ppmv, in orange RH values for 121 ppmv <WV < 2.5x121 ppmv and in red RH values for WV>2.5x121 ppmv. On the two panels, the black thick line corresponds to the monthly mean RH in the upper-troposphere.

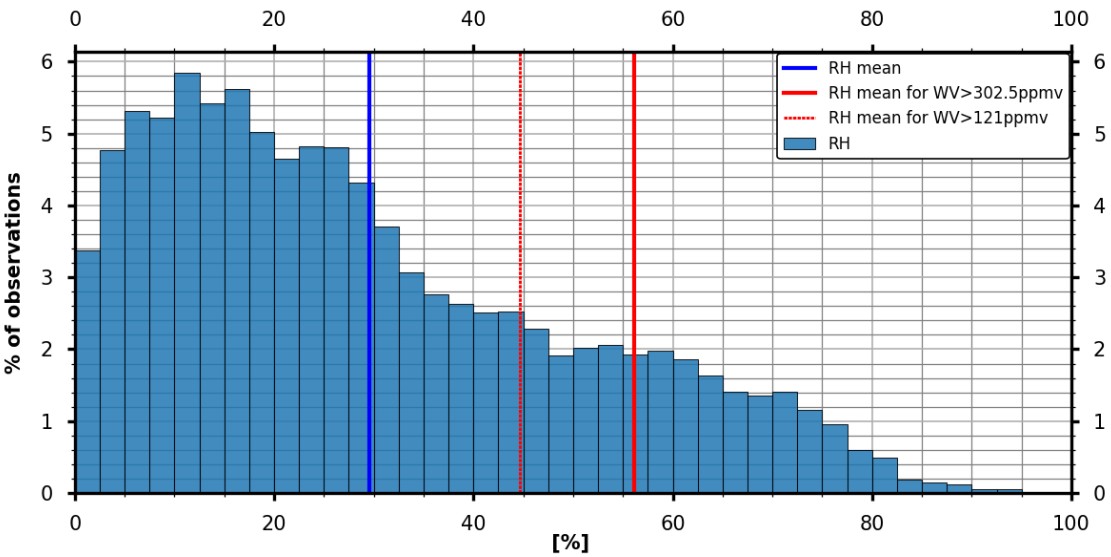

**Figure 5: RH distribution in the upper troposphere (10-13km) above Réunion Island. 904 profiles for the period Nov 2013-Apr 2016 have been used to compute the distribution. Bins are every 5%. The mean RH of the distribution (30.4%) is shown with a blue line, the mean RH for hydrated profile (WV>121ppmv) with a dotted red line and for strong hydration (WV>2.5x121ppmv) with a red line.**

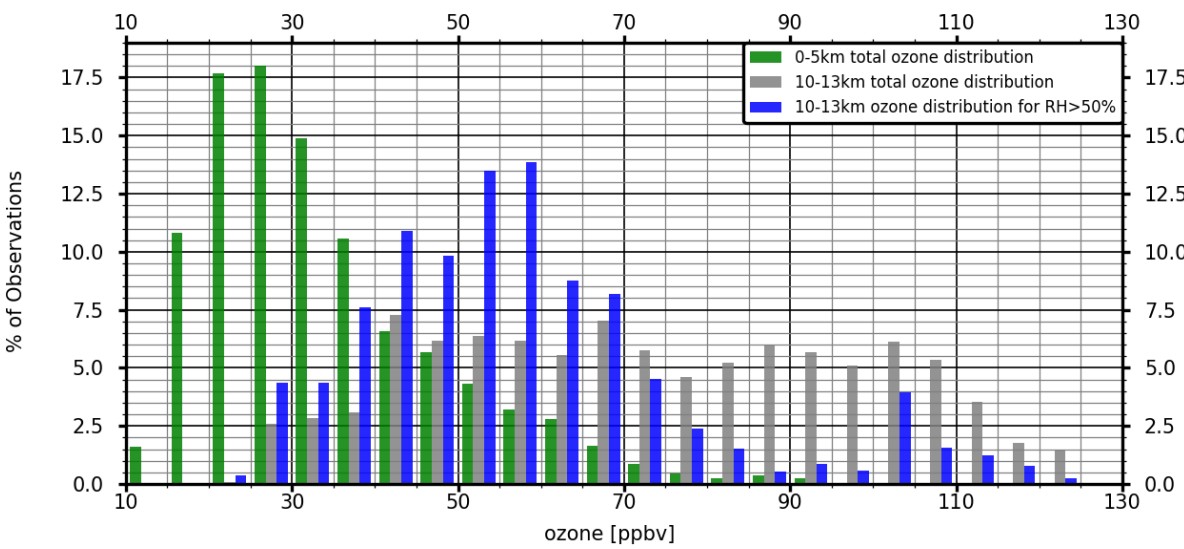

**Figure 6: NDACC/SHADOZ ozone distribution in the lower troposphere (0-5 km, in green). The total distribution in the upper troposphere (10-13km, in grey) and for moist data (10-13km and RH>50%, in blue). The distributions are based on 55 ozone profiles for austral summers 2014, 2015 and 2016. The mean for each distribution is 73.8 ppbv, 57 ppbv, 33.5 ppbv for the 10-13 total ozone distribution, 10-13 km ozone distribution with RH>50% and 0-5km total ozone distribution respectively.**

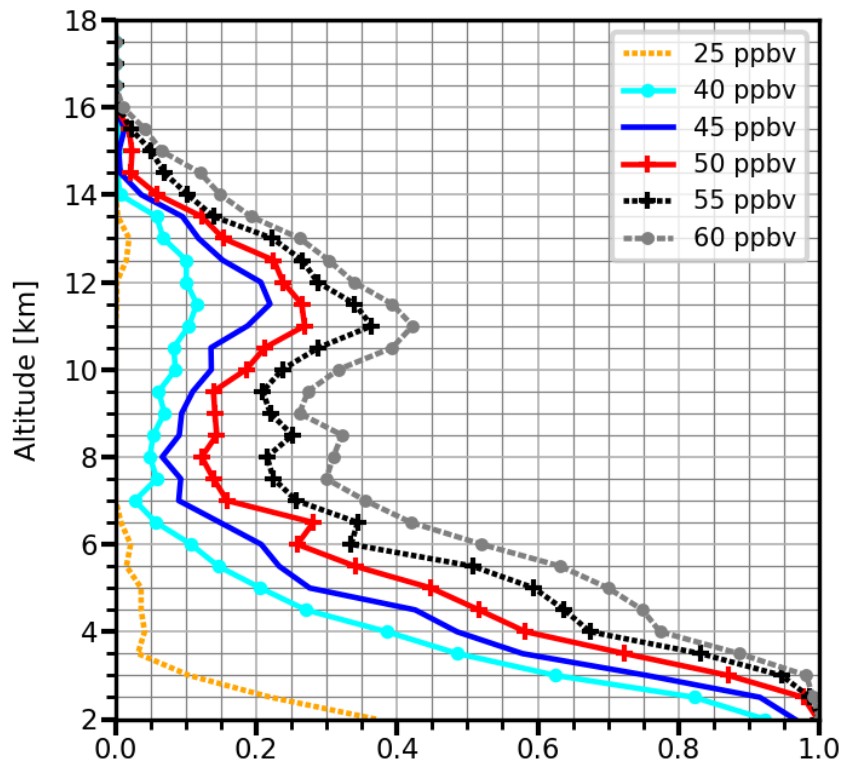

630

**Figure 7: Vertical profiles of frequency of occurrence of ozone mixing ratios below 25, 40, 45, 50, 55 and 60 ppbv at Réunion Island for austral summers (NDJFMA) 2014, 2015 and 2016**

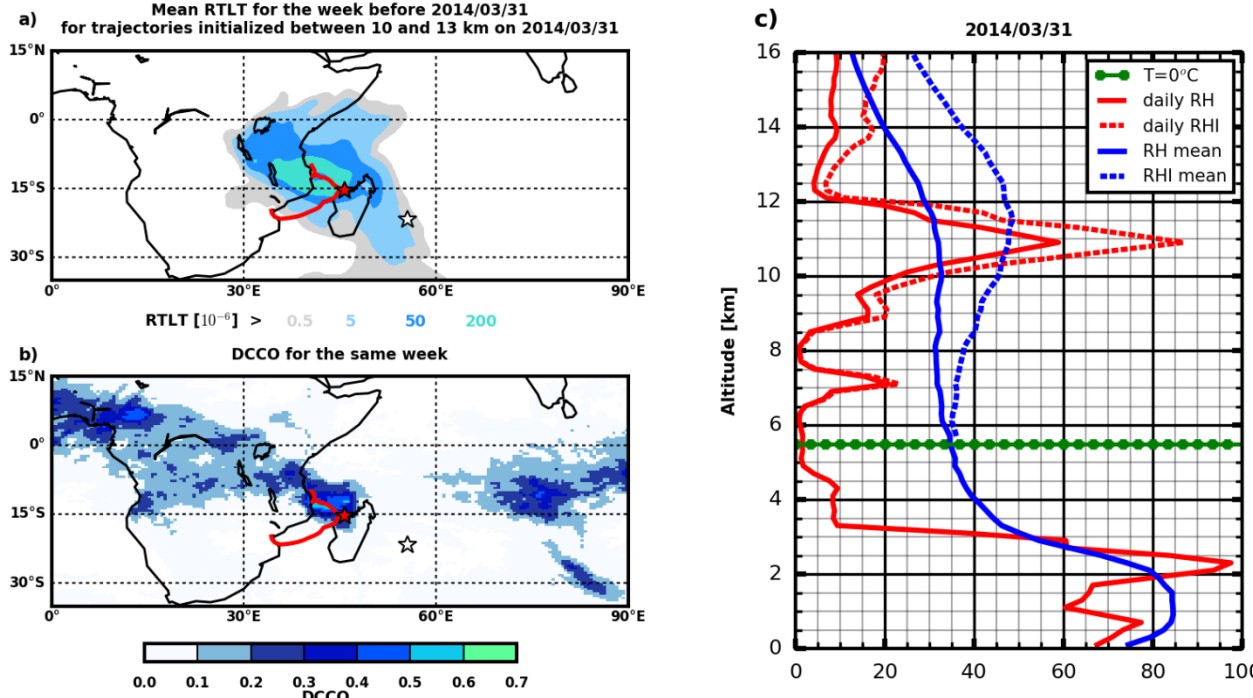

**Figure 8:**

a) (top-left) fraction of residence time below 5km (RTLT) for air masses initialized in the upper troposphere (10-13km) over Réunion Island on 31 March 2014.

b) (bottom-left) Map of DCCO for the week before 31 March 2014. On (a) and (b) the best-track data are provided by Météo-France. On the two panels on the left, the red star indicates TC Hellen position on 31 March 2014 and the white star corresponds to the location of Réunion Island

c) (right) RH/RHi profiles on 31 March 2014 (solid and dotted red lines respectively) and seasonal mean RH/RHi profiles (solid and dotted blue lines respectively) over austral summers 2014, 2015 and 2016. The dotted green line corresponds to the 0°C isotherm.

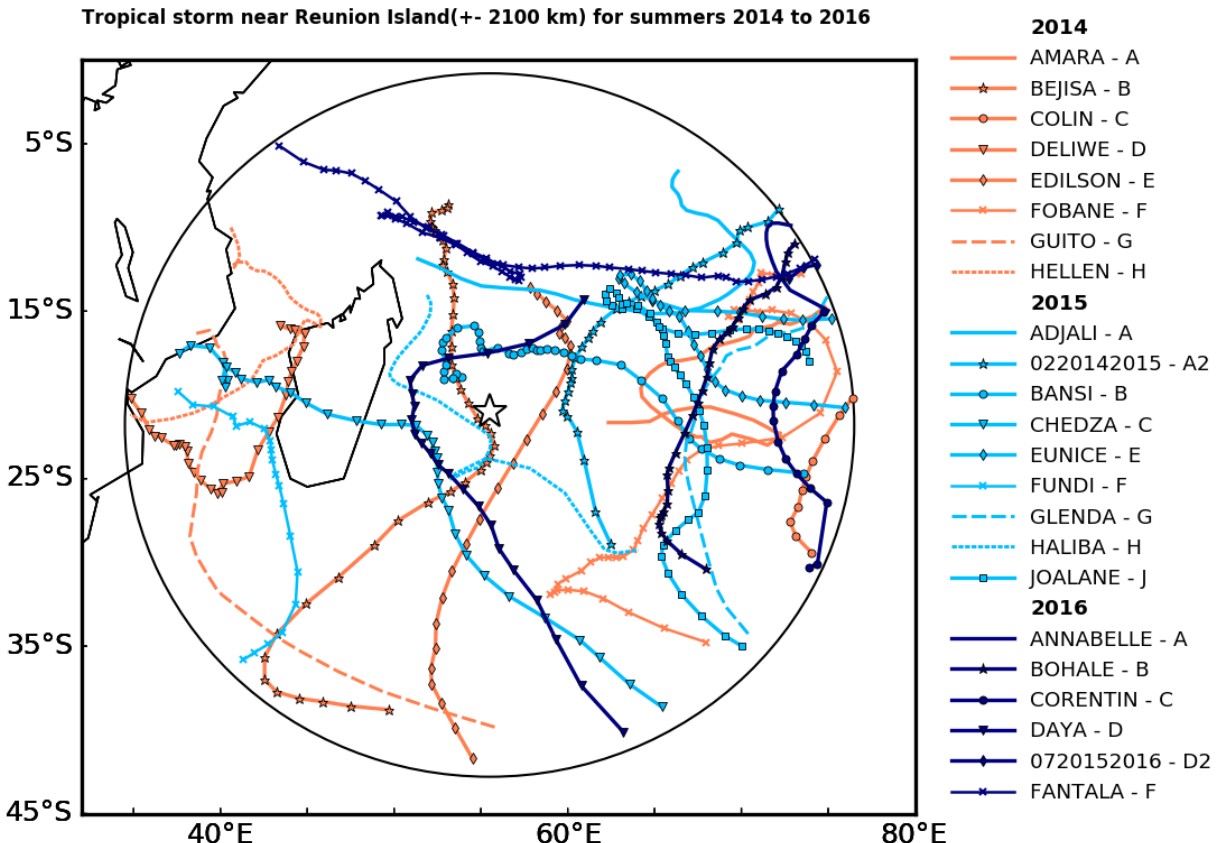

**Tropical storm near Reunion Island(+- 2100 km) for summers 2014 to 2016**

**2014**
AMARA - A
BEJISA - B
COLIN - C
DELIWE - D
EDILSON - E
FOBANE - F
GUITO - G
HELLEN - H
**2015**
ADJALI - A
0220142015 - A2
BANSI - B
CHEDZA - C
EUNICE - E
FUNDI - F
GLENDA - G
HALIBA - H
JOALANE - J
**2016**
ANNABELLE - A
BOHALE - B
CORENTIN - C
DAYA - D
0720152016 - D2
FANTALA - F

**Figure 9: Map of Tropical Cyclone's best-tracks for a range ring of 2100 km around Réunion Island. The best-tracks of tropical Cyclones during austral summer 2014 (November 2013 to April 2014) are shown in orange with different symbols for each TC, best-tracks for austral summer 2015 (November 2014-April 2015) are in cyan and in blue for austral summer 2016 (November 2015-April 2016, in green). The yellow star indicates the location of Réunion Island.**

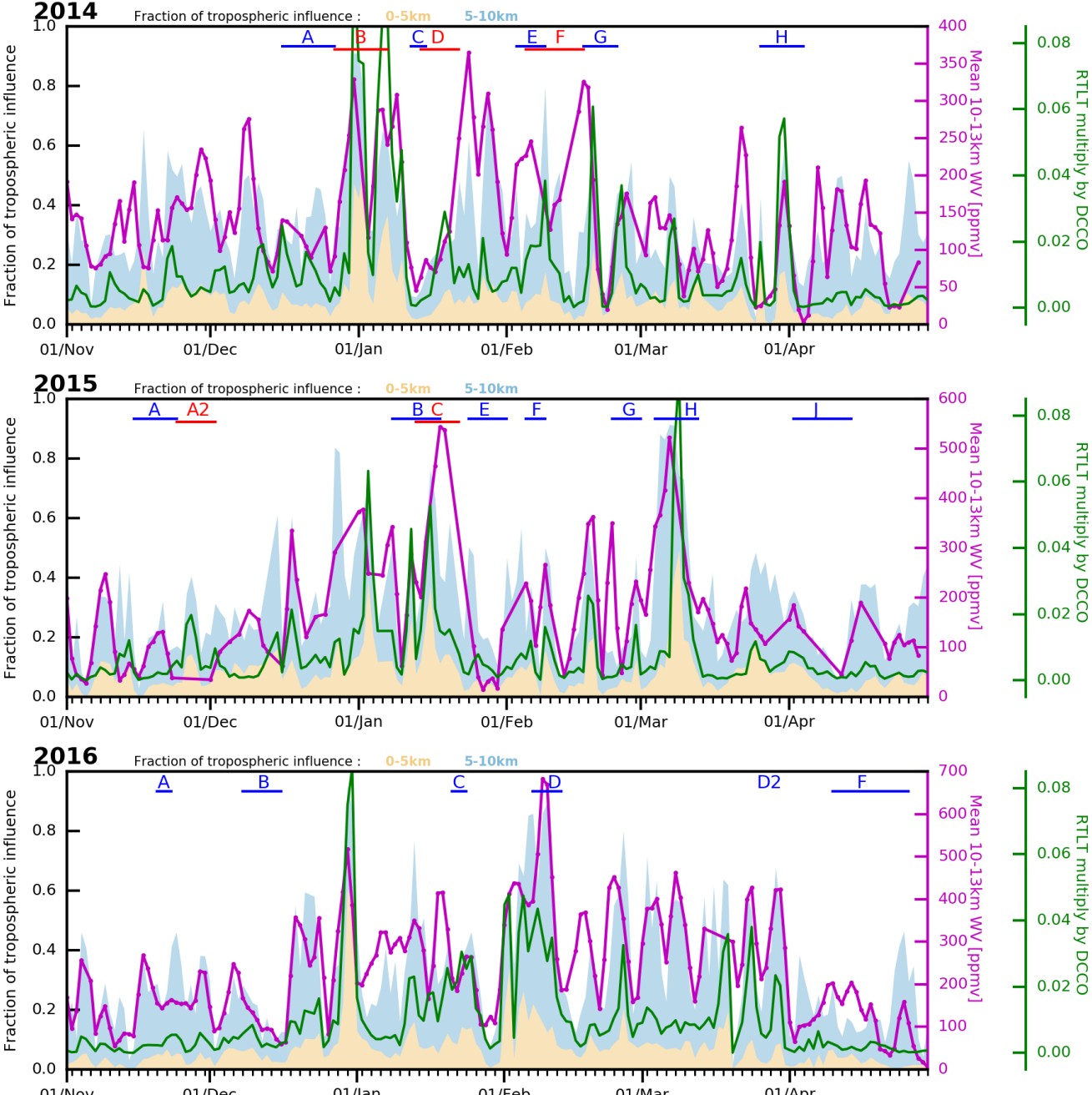

Figure 10: Top panel, Time-evolution of fraction of tropospheric influence (spatially summed) for the lower troposphere (0-5km, sRTLT) in orange and the middle troposphere (5-10km, RTMT) in blue. The product of DCCO by RTLT in green line and mean upper-tropospheric (10-13km) water vapor mixing ratio in purple for austral summer 2014. Middle and bottom panels: Same as top panel for austral summers 2015 and 2016 respectively. The letters in red and blue indicate periods with different tropical cyclones for each austral summer as shown on Figure 9.

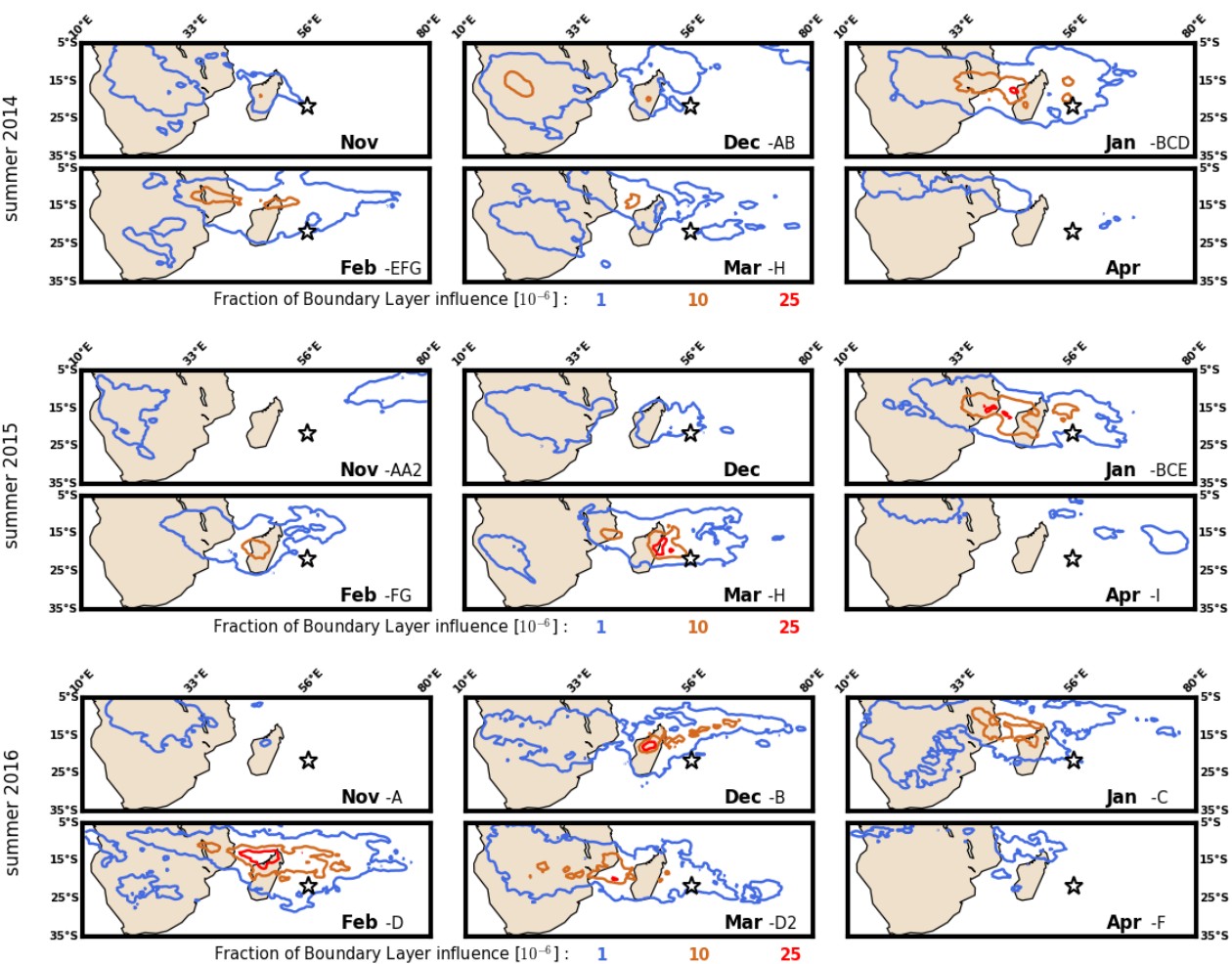

**Figure 11: Monthly average product of DCCO by RTLT normalized by the total residence time in the whole atmospheric column. From top right to bottom left: November 2013, December 2013, January 2014, February 2014, March 2014, April 2014, November 2014, December 2014, January 2015, February 2015, March 2015 and April 2015. The values for each contour are indicated by the numbers in blue, orange and red at the bottom of each panel. The location of Réunion Island is indicated by a white star on each plot. On each subplot, the letters indicate tropical cyclones that occurred during a given month.**

660