# Peer review of "Impact of convection on the upper-tropospheric composition (water vapor/ozone) over a subtropical site (Réunion Island, 21.1°S-55.5°E) in the Indian Ocean"

_Atmospheric Chemistry and Physics, 2020_

## Referee Comment (RC1) · Anonymous Referee #1 · 10 Mar 2020

**1   Content**

Ozone measurement within the SHADOZ network and humidity measurements from the years 2014-2016 are used to show the convective hydration of the upper troposphere in the vicinity of Reunion Island. Trajectory calculations in combination with METEOSAT brightness temperature are used to confirm the convective influence. Lower ozone mixing ratios are found in humid air masses, which are shown to be originating from the boundary layer. Part of the hydration is associated with tropical cyclones

passing the area around the island.

**2   Overall impression and rating**

The overall impression of the manuscript is good in general. The manuscript is well structured and the text is mostly easy to understand. I agree with most of the interpretations and I think you are right that most parts of the humidity can be linked to convection. The measurements and the message of the manuscript is a good contribution to the community. However, I have one concern of handling the relative humidity profiles with respect to clouds (see below). For these reasons, I recommend publication in ACP after major revisions.

**3   Major comment:**

My main concern in this study is the handling of RH observations itself and with respect to clouds. Cirrus clouds is the prevailing cloud type in the altitude range used in this study. These clouds are often found in the outflow of convective systems (e.g. Fierli et al, 2008) but they occur also due to other dynamical situations like warm or cold fronts, gravity waves etc. which produces also a slow to moderate uplift of air (see Kraemer et al, 2016). Clouds are always an indication of uplifted moisture, but they also redistribute water vapor to lower altitudes due to sedimentation of ice particles. Therefore, they can also partly weaken the hydration signatures in your observations.

In the whole study, all humidity measurements are shown as relative humidity with respect to water (I guess) which is the standard output of meteorological radiosondes. In the used altitude region (10-13 km) relative humidity with respect to ice would be the better quantity to identify cirrus clouds. For example, on page 5 (lines 161-166) the

60% RH (with respect to water) corresponds to 100% RH with respect to ice (RHi) at assumed temperature of 220K (higher RHi with even lower temperatures). RHi values around 100% are an indication for clouds. And I think there are a lot of cloudy profiles in the measurements, as you can see for example in Figure 3 or 4.

Figure 5 shows the distribution with all RH measurements between 10 and 13 Kilometers. The measurements between RH of 50 to 60% which corresponds approximately to 90-100% RHi are most likely in cloud observations of cirrus. This could be discernible by a small increase in the number of observations between 50 to 60%. Since, the most frequent Rhi is expected to be around 100% RHi due to thermodynamical reasons (see Kraemer et al, 2016). Another example is visible in Figure 4 where a RH of 90% (April 2015) is shown, which would correspond to RHi of 136-167% for temperatures of 230-200K.

If there is a significant amount of cloudy profiles within you measurements, I'm sure that there is also a part of it not generated by convection and formed by other uplifting prozesses (examples see above). These uplifting prozesses just transports air masses more from the middle troposphere into the upper troposphere and not from the boundary layer. This can be also seen by Figure 10. There are a lot of signatures in the mean relative humidity where no indicator either sRTLT or the product of RTLT and DCCO create a coincident signature. This could also partly explain the higher ozone concentration found in moist profiles in the upper troposphere compared to other studies, because the air is just originating from higher altitudes above the boundary layer with a higher ozone concentration.

Another point, is the usage of relative humidity. The RH as well as RHi depends beside water vapor also strongly on temperature. If you see an increase in relative humidity it can also be due to a temperature decrease. Therefore, I suggest to show in Figure 4, a seconds panel with the water vapor mixing ratios between 10-13 km instead of RH to confirm that the summertime increase in RH is due to an increase in the water vapor concentration and not only due to a temperature artifact/anomaly.

In general, I think it is correct to use RH as an indication of hydration, but at least there must be more discussion about the usage of RH like the dependence on temperature. I still agree with the main message of the study. However, I think that the humidity measurements must be discussed in a more balanced way including also a short discussion about the effect of clouds also in combination with Figure 10.

**4 Specific comments/questions:**

- Page 1, line 31: Maybe add here Riese et al. 2012. They showed nicely how changes in ozone and water vapor due to mixing processes change the radiative budget of the UTLS region.

- Page 2, line 48: "Marine boundary layer to the upper troposphere (Jorge ozone chemical lifetime is on the order of 50 days,". Can you please specify where you have the information from and for which altitude the stated lifetime of ozone is valid. Usually you have a strong increase of ozone lifetime with altitude in the troposphere.

- Page 3, lines 81-82: It would be consistent and also helpful to also report about precision and uncertainties of radiosonde measurements of the Meteomodem M10 sonde. In particular, the uncertainties of the humidity sensor would be helpful for the interpretation of the measurements.

- Page 4, lines 124-126: Can you state something about how frequent you detect the anvil outflow instead of the anvils. Maybe you have a rough estimate. Also I'm wondering about in situ formed cirrus clouds which are typically colder than 230K. Are they not detected by the Meteosat 7 SEVIRI instrument? If the they are detected , they would also strongly distort your statistics of DCCO. Can you please comment on this also.

- Page 4, line 130: What is actually meant by "backtrajectories were calculated every hour at 0.25° resolution...." ? The resolution of the wind field is already mentioned in the text. And can you please also add the vertical resolution in altitude range of interest. Please add also information about the temporal length of the backward trajectory calculation.

- Page 4, lines 130-132: Are you sure that the ECMWF forecast data are used for the backward trajectory calculation and not the analysis data? The deviation between forecast and analysis data could be large. The analysis data would better represent the real meteorological fields.

- Page 5, 172-174 in combination with Figure 4: It would be helpful, if you could include the monthly mean as a time series in the plot (e.g. black line). Because of the strong scatter, it is difficult to see where the mean value of each period would be. With a monthly mean RH climatology in addition to the scatter, it would be easier to see.

- Page 6, 192-195 in combination with Figure 5: It would be good to discuss the measurements in cloud at this point. Especially the increase in the distribution between 50-60% RH. Additionally, a the line which marks approximately the 100% RHi in Figure 5 would help to see which observations could be potentially effected by clouds. All measurements strongly above 60% RH could be potentially clear sky observations again, because in clouds the humidity is diminished by the diffusional growth of ice particles to the thermodynamical/dynamical equilibrium around 100% or slightly above.

- Page 6, lines 200-220: The process of mixing typically depends on the dynamics (e.g. wind shears, wave breaking etc.) but also on time as you wrote. I suggest to include a plot or at least some numbers how long the air masses stayed in the upper troposphere after the convective uplift and the time of the measurement. This would give some indication if the duration is on the typical timescale of mixing.

- Page 8, lines 252-253: I agree, that the signature in the RH profiles around 11km could be potentially from the cyclone. But I'm also sure that there is also a cloud layer at 11 km altitude. Because the RHi is around 100% (RH eq. 60%) and the decrease in the humidity below the layer is slower than above the layer which indicates the hydration effect due to sedimented ice crystals. Here, I would suggest a bit more discussion about that.

- Page 9, lines 299-305 in combination with Figure 10: There are many peaks in RH which do not correspond to any signature in both trajectory products (e.g. around 01/Dec 2015, End of Jan 2015, beginning of summer 2015/2016 and many more). It would be good, if there would be more discussion in the text to explain those peaks and possible reasons (see also my main comment above).

- Page 9: In Section 3.4.3 the geographical origin of convection is discussed with the help of the product of DCCO and RTLT. In text the discussion is linked to the respective cyclones. It would really be helpful to follow the discussion, if the names or at least the letters of the occurring cyclones (A-F) are written in each subplot of Figure 11. For example for March 2015 G and H or the full names of the cyclones.

**5   Technical comments/suggestions:**

- Page 4, line 123: period of "the" study

- Page 4, line 126: to our treatment "of" convective

- Page 5, lines 141-142: ". Residence times are computed using the model gridded output domain (1°x1° grid cells) values combining the results of the 8 3-hourly runs to provide a daily estimate of the source regions for air particles." This sentence is difficult to understand. Can you please rephrase.

- Page 5, line 146 and 147: "low" should be "lower" ?

- Page 8, line 269: Skip the word "was"

- Page 9, lines 288-290: Sentence needs to rephrased " estimated by correspondes ".

- Please check the capitalization of the word Figure in text (e.g. page 8, line 273). It should be consistent throughout the manuscript.

- Figure 8: It would be good, if you could include the location of Reunion island in the maps.

**6 References:**

- Fierli, F., Di Donfrancesco, G., Cairo, F., Marécal, V., Zampieri, M., Orlandi, E., and Durry, G.: Variability of cirrus clouds in a convective outflow during the Hibiscus campaign, Atmos. Chem. Phys., 8, 4547–4558, https://doi.org/10.5194/acp-8-4547-2008, 2008.

- Krämer, M., Rolf, C., Luebke, A., Afchine, A., Spelten, N., Costa, A., Meyer, J., Zöger, M., Smith, J., Herman, R. L., Buchholz, B., Ebert, V., Baumgardner, D., Borrmann, S., Klingebiel, M., and Avallone, L.: A microphysics guide to cirrus clouds – Part 1: Cirrus types, Atmos. Chem. Phys., 16, 3463–3483, https://doi.org/10.5194/acp-16-3463-2016, 2016.

- Riese, M., F. Ploeger, A. Rap, B. Vogel, P. Konopka, M. Dameris, and P. Forster, Impact of uncertainties in atmospheric mixing on simulated UTLS composition and related radiative effects, J. Geophys. Res., 117, D16305, doi:10.1029/2012JD017751, 2012.

---

## Referee Comment (RC2) · Anonymous Referee #3 · 17 Mar 2020

The paper presents an analysis of tropical upper tropospheric ozone and relative humidity based on 3 years of observations over Reunion Island, located in the Southern subtropics. Using backtrajectories from FLEXPART, the influence of deep convection is examined. The analyses focus on austral summer events of low ozone and high relative humidity, which are linked in several cases to air transported from the boundary layer by tropical cyclones passing within a radius of 2100 km around the island.

Overall the study presents novel analyses of quality data which provide interesting insights on the variability of humidity and ozone in the subtropical upper troposphere, it

is well written and the information is presented in a clear manner. Below I list my minor comments and technical corrections.

Minor comments:

- The authors could put their results a bit more into context, in particular when describing the seasonality of humidity and ozone in the subtropical region. For instance, the dry values in JJA as compared to DJF are connected to the ITCZ movement. Similarly, for the ozone only biomass burning is considered to explain the seasonality. But is there a role of other mechanisms such as transport from the stratosphere or from middle latitudes? This seems to be the case for the high ozone – low water vapour layers, which are ubiquitous over Reunion Island in JJA.

- Given the important role of the MJO for deep convection over the Indian Ocean, its possible influence on convective activity deserves more discussion: you could at least identify the phases in the period under study and try to establish a connection.

Technical corrections:

- L13: verb tense concordance: in general you use past tense, but some times present. This should be homogenized. For example here "are analyzed" should be "were analyzed"-

- L168-170 and L175-176: These two sentences are almost exactly the same, no need to repeat.

- L 183: remove 'several'

- L 184: 'Correlated' should be 'Consistent'?

- L185: The outflow from two tropical cyclones

- L186-189: This was already said before (L172-174)

- L203: tropical free troposphere

- L219: in the tropical marine. . .

- L225: In the Solomon et al. Study. . .

- L231: This explains. . .

- L250: Remove comma after Although

- L268: was the advected eastward

- L289: estimated by FLEXPART

- L 293 Swap letters G and H

- L293: missing letters C and D
* * *

---

## Referee Comment (RC3) · Anonymous Referee #2 · 25 Mar 2020

The authors present a 3-year analysis of relative humidity (RH) and ozone in the upper troposphere at a tropical location removed from frequent deep convection. It is convincingly shown that convection is likely responsible for significant reduction in upper troposphere ozone and enhancement in RH. The analyses are well described and will make a useful contribution to the literature. However, before it is suitable for publication, a few outstanding issues should be resolved.

Major Comments:

[Figure]

1. The RH data used comes from daily radiosonde measurements. The standard instruments used in radiosondes have long been known to suffer from significant dry biases in the upper troposphere and stratosphere due to instrument limitations and icing in supercooled liquid clouds (e.g., Miloshevich et al. 2004). No mention is given in the article on the quality of the RH measurements in the radiosonde data used and whether or not a correction has been applied to these data to account for known sources of dry bias. This is an important issue because it impacts much of the analysis presented.

2. The use of a trajectory model to track air mass history and identify boundary layer sources of air associated with convection is a good approach to analysis, but the accuracy of the parameterized sub grid-scale motions along the trajectories is not well demonstrated. How well does the parameterized convection match actual convection in this region? The reliability of this approach is fundamental to the analysis and arguments presented in the paper and an assessment should be provided. The lack of agreement in RTLT and DCCO for the case study included in the paper (i.e., Figure 8) is particularly concerning as it suggests the parameterized convection fails to represent much of that observed (at least for the week shown). The only element of the paper acknowledging this potential issue (lines 299-305) is, in my opinion, insufficient.

3. There is a missed opportunity to put case study of tropical cyclones into broader context. Previous work on the impact of tropical cyclones on upper troposphere and lower stratosphere water vapor and ozone is not acknowledged and would help in the authors' interpretation and argumentation here (e.g., Ray & Rosenlof, 2007; Zhan & Wang, 2012). Moreover, I do not find the impact of tropical cyclones on upper troposphere RH to be convincing in the paper, likely related to my concerns outlined in #2 above.

Specific Comments:

Lines 134-135: "lower upper-tropospheric ozone values observed" is a bit confusing

phrasing. Suggest rephrasing to "lower observed ozone values in the upper troposphere"

Line 144: suggest revising "the day" and "the latitude" to "day" and "latitude"

Line 183: "affected by several three tropical cyclone events" Which is it – several or three?

Lines 194-195: Why is a threshold of 50% chosen? What is the sensitivity to this choice?

Lines 219-221: This statement doesn't seem appropriate. What if the responsible convection is land-based? One should expect a higher ozone mixing ratio in the boundary layer in that case. It seems reasonable that many/most convective sources for air in the upper troposphere at Reunion island would be land-based (e.g., look at Figure 3!).

Line 221: comma should be a period

Lines 343-346: as presented, this seems anecdotal and based on a single case. It would be more convincing to show a map of the FLEXPART convective sources (i.e., locations of most recent position in the lower troposphere) for matches with the RH profiles. It would help to better answer the question of importance of differences in boundary layer sources and mixing to impacting the upper troposphere ozone observed.

References: Miloshevich et al. 2004: Development and Validation of a Time-Lag Correction for Vaisala Radiosonde Humidity Measurements, J. Atmos. Oceanic Technol., 21, 1305–1327

Ray & Rosenlof, 2007: Hydration of the upper troposphere by tropical cyclones, J. Geophys. Res., 112, D12311, doi:10.1029/2006JD008009

Zhan & Wang, 2012: Contribution of tropical cyclones to stratosphere-troposphere exchange over the northwest Pacific: Estimation based on AIRS satellite retrievals and ERA-Interim data, J. Geophys. Res., 117, D12112, doi:10.1029/2012JD017494

---

## Author Comment (AC1) · 12 May 2020

Please, find the response to the comments that you posted as well as the corrected manuscript in the supplement.

Please also note the supplement to this comment:
https://www.atmos-chem-phys-discuss.net/acp-2020-2/acp-2020-2-AC1-supplement.zip

---

## Author Response (AR1)

As suggest by the ACP review, the document is organized as follow, in chronological order:

-Answer to Anonymous Referee #1: pages 2 to 14

-Answer to Anonymous Referee #3: pages 14 to 16

-Answer to Anonymous Referee #2: pages 17 to 23

- Major changed in the manuscript:  pages 24 to 29

-New manuscript, with the added section in red at the end of the document.

**Anonymous Referee #1:**

We appreciate the thoughtful and constructive comments from the reviewers. Their helpful suggestions and attention to detail have made this a substantially better paper, and we greatly appreciate the time they put into the manuscript.

Please see below our responses (in bold) to the individual detailed comments. Numerous figures are shown in our response to illustrate our points but some are not included in the revised manuscript.

We have addressed all the reviewers 'comments and modified the manuscript and figures accordingly.

**Major comments:**

My main concern in this study is the handling of RH observations itself and with respect to clouds. Cirrus clouds is the prevailing cloud type in the altitude range used in this study. These clouds are often found in the outflow of convective systems (e.g. Fierli et al, 2008) but they occur also due to other dynamical situations like warm or cold fronts, gravity waves etc. which produces also a slow to moderate uplift of air (see Kraemer et al, 2016). Clouds are always an indication of uplifted moisture, but they also redistribute water vapor to lower altitudes due to sedimentation of ice particles. Therefore, they can also partly weaken the hydration signatures in your observations.

In the whole study, all humidity measurements are shown as relative humidity with respect to water (I guess) which is the standard output of meteorological radiosondes. In the used altitude region (10-13 km) relative humidity with respect to ice would be the better quantity to identify cirrus clouds. For example, on page 5 (lines 161-166) the 60% RH (with respect to water) corresponds to 100% RH with respect to ice (RHi) at assumed temperature of 220K (higher RHi with even lower temperatures). RHi values around 100% are an indication for clouds. And I think there are a lot of cloudy profiles in the measurements, as you can see for example in Figure 3 or 4.

**The radiosonde measurements presented in the first version of the manuscript correspond to relative humidity with respect to water (RH). For temperatures below 0°C, we have computed the RH with respect to ice (RHi) using the formula of Hyland and Wexler (1983) for saturation vapor pressure over ice. Figures 3 and 4 have been modified accordingly. Section 2.1 in the revised manuscript has been corrected.**

[Figure]

*Figure 1: Distribution of RH with respect to water (RH, blue) and RH with respect to ice (RHi, orange) in the upper troposphere (10-13km) above Réunion Island. 904 profiles for the period Nov 2013-Apr 2016 have been used to compute the distribution. Bins are every 5%. The blue thick line indicates the mean RH (30%) and the red thick line indicates the RHi mean (46.5%).*

Figure 1 above compares RH and RHi distributions between 10 and 13 km for the period Nov 2013-Apr 2016. As you suggested, there is a significant number of profiles with RHi above 80% (20% of the profiles) and above 100% (8.7 % of the profiles). We agree that RHi values ≥ 100% could be an indication of the presence of clouds. However additional remote-sensing instruments (e.g. Lidar/Radar) would be needed in addition to

the radiosonde measurements of RH to truly assess the presence of these clouds. Unfortunately, the daily Météo-France and weekly SHADOZ radiosondes are performed at the airport (Gillot Station, in the north of the island) where no remote sensing instruments are installed.

Figure 2 shows the daily evolution of mean upper-tropospheric (10-13km) RH (with respect to water) from September 2013 to July 2016. On Figure 2 the RH values are color-coded according to the RHi (top) and the water vapor mixing ratio (bottom) values. The different RHi/WV ranges used in Figure 2 correspond to our definition of dry profiles (in blue), wet profiles (orange) and supersaturated profiles (red). To distinguish the effect of temperature and water vapor on RH/RHi values, we computed the water vapor mixing ratio (WV) for each profile between September 2013 to July 2016. We found a mean 10-13km WV of 121ppmv over this period. Note that we compared this value with a climatological WV value using Microwave Limb Sounder (MLS) v4.2 water vapor data for 2005-2017. We computed a MLS climatological WV profile for a region of 5°x5° surrounding Réunion Island. The MLS climatological WV profile at 261 hPa (~10km) is 116 ppmv and this agrees with the mean upper-tropospheric value of 121 ppmv inferred from the radiosonde data.

The top panel on Figure 2 shows that 91% of profiles with RH>50% are associated with high RHi > 80%. These events have also a high WV (lower panel of Figure 2) and indicate a hydration rather than a cooling effect on the high RH/RHi values. Some hydrated profiles (RH > 50%, 9% of the profiles) with a low RHi (< 80%) are present in January 2016 and this could be linked to a cooling of the upper troposphere.

27% of the hydrated profiles (RH>50%) correspond to supersaturated RHi (RHi>100%) and occur mostly in in 2015 and 2016. The RH values associated with RHi>100% in 2015 are spread throughout the summer months. They correspond to air masses detrained from tropical cyclones (Bejisa in January 2015 and Haliba in March 2015) or have high resident time in the middle troposphere (RTMT 5-10km on Figures 3 &10 of the revised manuscript). In 2016, the ice saturated profiles are almost all between February and March. They are associated with a high RTMT (RTMT on Figure 3).

A discussion was added in section 3.1 in the revised manuscript. As the distinction between high RHi (>80%) and low RHi (<80%) is similar to the distinction between hydrated profiles (RH>50%) and dry profiles (RH<50%), we choose to keep and show RH (with respect to water) measurement to study convective effects on the hydration of the upper troposphere above Réunion Island.

[Figure]

*Figure 2: Top, daily evolution of mean upper tropospheric (10 to 13 km) RH for the period September 2013 to July 2016. The RH (with respect to water) values are color coded according to the values of RHi. In blue RH values for RHi <80%, in orange RH values for 80% <RHi < 100%, and in red RH values for RHi> 100%.*
*Bottom, daily evolution of mean upper tropospheric (10 to 13 km) RH for the period September 2013 to July 2016. The RH (with respect to water) values are color coded according to the values of water vapor mixing ratios (WV). The mean 10-13km WV is 121 ppmv. In blue RH values for WV < 121 ppmv, in orange RH values for 121 ppmv <WV < 2.5x121 ppmv and in red RH values for WV>2.5x121 ppmv.*
*On the two panels, the black thick line corresponds to the monthly mean RH in the upper-troposphere.*

[Figure]

*Figure 3: Top panel, Time-evolution of fraction of tropospheric influence (spatially summed) for the low troposphere (0-5km, sRTLT) in orange and the middle troposphere (5-10km, RTMT) in blue. The product of DCCO by RTLT in green line and mean upper-tropospheric (10-13km) water vapor mixing ratio in purple for austral summer 2014. Middle and bottom panels: Same as top panel for austral summers 2015 and 2016 respectively. The letters in red and blue indicate periods with different tropical cyclones for each austral summer as shown on Figure 9 of the manuscript.*

Figure 5 shows the distribution with all RH measurements between 10 and 13 Kilometers. The measurements between RH of 50 to 60% which corresponds approximately to 90-100% RHi are most likely in cloud observations of cirrus. This could be discernible by a small increase in the number of observations between 50 to 60%. Since, the most frequent RHi is expected to be around 100% RHi due to thermodynamic reasons (see Kraemer et al, 2016). Another example is visible in Figure 4 where a RH of 90% (April 2015) is shown, which would correspond to RHi of 136-167% for temperatures of 230-200K.

**As shown on the top panel of Figure 2, 96% of profiles with high RHi (>80%) correspond to RH >50%. There was only one case in 2016 with a high RHi and a RH of 45%. We agree that these high values could correspond to cirrus clouds in the upper troposphere. However, it is difficult to assess the presence of such clouds based on**

**RHi values only. Additional remote sensing data (e.g. Lidar/Cloud Radar) would be required but are not available for the Gillot Station where the radiosonde launches are performed. A comment has been added in section 3.1 of the revised manuscript.**
**The RH value of 87% on 7 March 2015 corresponds to a RHi of 137%. This high value is linked with the passage of tropical storm Haliba (March 7-10) near Réunion Island.**

If there is a significant amount of cloudy profiles within your measurements, I'm sure that there is also a part of it not generated by convection and formed by other uplifting processes (examples see above). These uplifting processes just transport air masses more from the middle troposphere into the upper troposphere and not from the boundary layer. This can be also seen by Figure 10. There are a lot of signatures in the mean relative humidity where no indicator either sRTLT or the product of RTLT and DCCO create a coincident signature. This could also partly explain the higher ozone concentration found in moist profiles in the upper troposphere compared to other studies, because the air is just originating from higher altitudes above the boundary layer with a higher ozone concentration.

**We have computed the resident time in middle troposphere (RTMT, 5-10km) in addition to the resident time in the low troposphere (RTLT, 0-5km) using the FLEXPART backtrajectories. The evolution of RTMT is shown in blue on Figure 3. In addition, we have added the evolution of the mean upper-tropospheric water vapor mixing ratio (purple line) instead of the mean upper-tropospheric RH (to distinguish between hydration versus cooling effect by uplift). The sum of RTLT and RTMT corresponds to the total residence time in the troposphere from 0 to 10km.**
**On Figure 3, a high correlation can be seen between the sum of RTLT and RTMT and mean upper-tropospheric WV variations. Therefore, hydration effect (mostly from convection) dominates over cooling effect due to uplift from the low/middle troposphere. The high correlation between RTLT+RTMT and the mean upper-tropospheric WV indicates that water vapor transport occurred either from the low troposphere (probably by deep convection) or from the middle troposphere. The transport of water vapor from the middle-troposphere could be due large-scale uplift of air masses as you suggested.**

**An analysis on the correlations between WV mixing ratio and residence time calculated by FLEXPART at different altitude range in the troposphere is shown on figure 4 below. The first row shows the correlation between WV and RTLT (0-5km, left) or the product of RTLTxDCCO (right). According to the correlation calculations, 23% to 27% of the WV variability is explained by RTLT or RTLTxDCCO. As you mentioned, different phenomena can influence the upper tropospheric water vapor variability, such as deep convection, gravity waves or large-scale uplift of air masses. The low value of the R² coefficient is therefore difficult to interpret.**

**The second row on Figure 4 shows the correlation between WV mixing ratio and the rate of residence time in the middle troposphere (RTMT) between 5-10km altitude. The R-squared coefficient between RTMT and WV is 42%. The study of Schumacher et al. (2015) has shown that stratiform clouds have a vertical speed up to 10 m s$^{-1}$ below 7km in altitude and then a slow ascent (< 0.5 m s$^{-1}$) up to 10km. It suggests that a higher correlation between WV and RTMT than between WV and RTLT can be expected for these kinds of clouds.**

**To conclude, the RTLTxDCCO product can represent only the influence of deep convective clouds, while higher R-squared coefficient with RTMT with WV shows that stratiform clouds can contribute to enhance the WV mixing ratio in the 10-13km range. Therefore the correlation (R-squared of 42%) between the sum of (RTLT+RTMT)xDCCO and the mean upper-tropospheric WV indicates water vapor transport associated with convection; either from the low troposphere by deep convective clouds or from the middle troposphere by stratiform clouds.**

[Figure]

*Figure 4: Correlations between mean upper-tropospheric water vapor mixing ratio (WV), residence times in the troposphere (0-5km RTLT, 5-10km RTMT and 0-10km RTLT+RTMT) and tropospheric residence times multiplied by DCCO. Backtrajectories over one-week were computed with the FLEXPART Lagrangian model.*

Another point is the usage of relative humidity. The RH as well as RHi depends beside water vapor also strongly on temperature. If you see an increase in relative humidity it can also be due to a temperature decrease. Therefore, I suggest to show in Figure 4,a seconds panel with the water vapor mixing ratios between 10-13 km instead of RH to confirm that the summertime increase in RH is due to an increase in the water vapor concentration and not only due to a temperature artifact/anomaly.

**We propose to add Figure 2 above to the revised manuscript. On this figure, the link between high water vapor mixing ratios and mean-upper tropospheric RH can be seen. The red crosses indicate that the most hydrated profiles (2.5 x the mean upper-tropospheric of 121ppmv over Sept 2013-Mar 2016) are associated with RH>50%. There are few events (~5 profiles) with WV > 302.5 ppmv and a low RH (<50%). They occur mostly in austral summer 2016. These events are not discussed on the paper but could be linked to warmer air from recent convection.**

In general, I think it is correct to use RH as an indication of hydration, but at least there must be more discussion about the usage of RH like the dependence on temperature. I still agree with the main message of the study. However, I think that the humidity measurements must be discussed in a more balanced way including also a short discussion about the effect of clouds also in combination with Figure 10.

**As discussed above, there is a high correlation with high RHi/RH values and mean upper-tropospheric water vapor mixing ratio. Therefore, we choose to keep RH (with respect to water) measurements to study convective effects on the hydration of the upper troposphere above Réunion Island. We did show some RHi values on Figures 3&8 of the revised manuscript.**

**Specific comments/questions:**

• Page 1, line 31: Maybe add here Riese et al. 2012. They showed nicely how changes in ozone and water vapor due to mixing processes change the radiative budget of the UTLS region.
**The reference has been added.**

• Page 2, line 48: "Marine boundary layer to the upper troposphere (Jorge ozone chemical lifetime is on the order of 50 days,". Can you please specify where you have the information from and for which altitude the stated lifetime of ozone is valid. Usually you have a strong increase of ozone lifetime with altitude in the troposphere.
**The reference is at the end of the sentence, the study of Folkins et al. (2002) has shown this result.**

• Page 3, lines 81-82: It would be consistent and also helpful to also report about precision and uncertainties of radiosonde measurements of the Meteomodem M10 sonde. In particular, the uncertainties of the humidity sensor would be helpful for the interpretation of the measurements.

**We compared the M10 measurements of RH (with respect to water) with Cryogenic Frospoint Hygrometer (CFH) water vapor sondes at the Maïdo Observatory (21.08°S, 55.38°E on the west coast of the island, 20 km away from the airport). Balloon-borne measurements of water vapor and temperature started in 2014 at the Maïdo Observatory on a campaign basis within the framework of the *Global* Climate Observing System (GCOS) *Reference* Upper-Air Network (*GRUAN*) network (Bodeker et al., 2015). The balloon sonde payload consists of the CFH and the Intermet iMet-1-RSB radiosonde for data transmission. The iMet-1-RSB radiosonde provides measurements of pressure, temperature, Relative Humidity (RH) and wind data (speed and direction from which zonal and meridional winds are derived). The CFH was developed to provide highly accurate water vapor measurements in the Tropical Tropopause Layer (TTL) and stratosphere where the water vapor mixing ratios are extremely low (~2 ppmv). CFH mixing ratio measurement uncertainty ranges from 5% in the tropical lower troposphere to less than 10% in the stratosphere (Vömel et al., 2007); a recent study shows that the uncertainty in the stratosphere can be as low as 2-3% (Vömel et al., 2016). The CFH instrument is often launched in tandem with the Modem M10 sonde. The CFH RH data were calculated with the CFH water vapor mixing ratio and the Intermet iMet-1-RSB temperature using the water vapor pressure equation by Hyland and Wexler (1983) and interpolated to the same 200-m vertical grid as the M10 data. A total of 17 multiple-payload (CFH+M10) soundings is used for the comparison shown on Figure 5. The RH profiles from the CFH and Modem M10 show good agreement with differences of less than 10% mostly from the surface to the stratosphere. In the lower troposphere, below 5km, the mean RH difference is -1.05%. In the middle (5-10km) and upper troposphere (10-15km) the mean RH differences are 1.5 and 2.2% respectively. Near 15km, the M10 RH shows dry biases with a peak difference of -3.7% at 15.6km (the mean RH at this level ranges from 18.5 for the M10 sonde to 22.2% for the CFH sonde). Figure 5 is not shown in the revised manuscript but we have included some text on the comparison between the M10 and CFH sondes in section 2.1 of the revised manuscript.**

[Figure]

**Figure 5:** *Vertical profiles of mean RH obtained from 17 multiple-payload sounding of CFH&M10 radiosondes (blue and black curves respectively on the left panel) at the Maïdo Observatory over 5 years (2014-2019). The mean profile of differences in RH is shown on the right panel.*

Page 4, lines 124-126: Can you state something about how frequent you detect the anvil outflow instead of the anvils. Maybe you have a rough estimate. Also, I'm wondering about in situ formed cirrus clouds which are typically colder than 230K. Are they not detected by the Meteosat 7 SEVIRI instrument? If they are detected, they would also strongly distort your statistics of DCCO. Can you please comment on this also?

**The infrared channel of the Meteosat Visible and Infra-Red Imager (MVIRI) instrument is centered at 11.5µm (between 10.5µm and 12.5µm). To our knowledge, there is no study on in situ formed cirrus clouds using the MVIRI instrument. However, Minnis et al. (2008) used the Moderate Resolution Imaging Spectroradiometer (MODIS) 11µm IR channel data and data taken by the Cloud-Aerosol Lidar and Infrared Pathfinder Satellite Observations (CALIPSO) to investigate the difference between cloud-top altitude $Z_{top}$ and infrared effective radiating height $Z_{eff}$ for optically thick ice cloud (i.e. deep convective clouds). From the MODIS IR brightness temperature data ($T_b$), they estimated cloud top altitudes by finding the altitude where $T_b$ is found in an atmospheric temperature sounding. The derived cloud top altitudes were then compared to cloud top altitudes estimated with CALIPSO observations. The difference $\Delta Z$ between $Z_{top}$ and $Z_{eff}$ rises from ~1.25 km for $Z_{eff}$ = 5 km to more than 2 km for $Z_{eff}$ > 14 km. This suggests than using a threshold of 230K to define deep convective clouds can induce an error in the selection of these clouds. Thin cirrus clouds could be included in our selection of deep clouds but it is difficult to say how much by using passive satellite sensors only. Additional measurements from active sensors such as CALIPSO would be required to distinguish between the deep convective cores inferred from passive infrared radiances and cold in situ formed cirrus clouds. However, this is beyond the scope of this study.**
**We did try to assess the sensitivity of the DCCO distribution to different $T_b$ thresholds. This is shown on Figure 6 below. We used 3 values of $T_b$: 220, 230 and 240 K that correspond to cloud top altitudes of 13.5km, 12km and 11km respectively. As shown by Minnis et al. (2008), the cloud top altitude estimated from passive infrared**

[Figure]

*Figure 6: Average map of the deepest convective cloud occurrence (DCCO) for austral summer conditions (NDJFMA from November 2013 to April 2016) using different brightness temperature thresholds (top left 220K, center left 230K, bottom left 240K). The yellow contour is for DCCO > 7%, the green contour for DCCO > 12% and the dark green contour is for DCCO > 17%. On the right, a temperature profile averaged over Austral Summers 2014, 2015 and 2016 is shown to illustrate the corresponding cloud top altitude.*

radiances are underestimated by 1 to 2km thus we added 1km to the altitude where $T_b$ is found in the mean temperature profile for austral summer conditions.

The distribution of DCCO using the 220K threshold is more selective and highlight the deeper convective clouds over Africa and the north of Madagascar. This threshold seems to be more adapted to identify deep convective centers associated with convection over land. As the $T_b$ threshold is increased to 230 and 240K more clouds over the ocean are included in the distribution of DCCO.

Young et al. (2013) have studied MODIS cloud top brightness temperature data in the tropics. They found that the averaged brightness temperature for deep clouds was around 228 K and 236.3K for anvil. Therefore, our definition of DCCO with a $T_b$ of 230K is a good compromise to distinguish deep convective clouds. Some cirrus clouds may be included in this distribution but it is hard to conclude without active remote sensor data. A comment was added in section 2.2 of the revised manuscript.

Page 4, line 130: What is actually meant by "backtrajectories were calculated every hour at 0.25∘resolution...." ? The resolution of the wind field is already mentioned in the text. And can you please also add the vertical resolution in the altitude range of interest. Please also add information about the temporal length of the backward trajectory calculation.

**We apologize for the confusion. FLEXPART can provide gridded output of the residence time. We ran backward simulations over 2 weeks and the residence time field was reported on a regular 0.5°x0.50° output grid every 3 hours. The resolution of the gridded output is totally independent from those of the meteorological input. Therefore, we use 0.25°x0.25° operational ECMWF input fields to compute the backward trajectories and the resulting residence time is reported on a regular 0.5°x0.5° output grid. We use input meteorological**

fields from the ECMWF Integrated Forecast System (IFS, current ECMWF operational data) that have 137 vertical levels up to 0.01hPa. The vertical resolution varies from ~20 m near the surface, ~100 m in the low troposphere, ~300 m in the middle/upper troposphere and 500 m in the stratosphere. Section 2.3 has been corrected in the revised manuscript.

• Page 4, lines 130-132: Are you sure that the ECMWF forecast data are used for the backward trajectory calculation and not the analysis data? The deviation between forecast and analysis data could be large. The analysis data would better represent the real meteorological fields.

**Once again, we apologize for the confusion. Operational data from the ECMWF IFS are used as input data for the FLEXPART simulations. FLEXPART was driven by using operational ECMWF analysis at 00, 06, 12 & 18 UTC and the 3 & 9-hour forecast fields from the 00 and 12 UTC model analysis. Section 2.3 has been corrected in the revised manuscript.**

• Page 5, 172-174 in combination with Figure 4: It would be helpful, if you could include the monthly mean as a time series in the plot (e.g. black line). Because Of the strong scatter, it is difficult to see where the mean value of each period would be. With a monthly mean RH climatology in addition to the scatter, it would be easier to see.

**The monthly mean value of RH has been added on Figure 2 of the revised manuscript.**

Page 6, 192-195 in combination with Figure 5: It would be good to discuss the measurements in the cloud at this point. Especially the increase in the distribution between 50-60% RH. Additionally, a line which marks approximately the 100% RHi in Figure 5 would help to see which observations could be potentially affected by clouds. All measurements strongly above 60% RH could be potentially clear sky observations again, because in clouds the humidity is diminished by the diffusional growth of ice particles to the thermodynamic/dynamical equilibrium around 100% or slightly above.

**As previously mentioned in our response, to distinguish the effect of temperature and water vapor on RH/RHi values, we computed the water vapor mixing ratio (WV) for each profile between September 2013 to July 2016. We found a mean upper tropospheric (10-13km) WV of 121 ppmv over this period. We use this threshold in Figure 5 of the revised manuscript to add 2 lines indicating the mean RH values for "hydrated profiles" with WV mixing ratios > 121 ppmv (RH ~45%) and "strongly hydrated" profiles for WV mixing ratios >2.5x121ppmv (RH ~56%) with a red line. We think this is a better alternative to showing the 100% RHi as we only include the hydration effect on the mean RH values rather than the temperature effect. These lines help the justification of the threshold of RH=50% to characterize hydrated air masses (the sensitivity of this threshold is discussed in our response to reviewer 3). As you suggested, we have added some discussion in the revised manuscript on the possible presence of clouds for RH between 50&60%.**

• Page 6, lines 200-220: The process of mixing typically depends on the dynamics (e.g. wind shears, wave breaking etc.) but also on time as you wrote. I suggest to include a plot or at least some numbers how long the air masses stayed in the upper troposphere after the convective uplift and the time of the measurement. This would give some indication if the duration is on the typical timescale of mixing.

**We have computed the residence time in the middle troposphere (RTMT, 5-10km) using the FLEXPART backtrajectories and the evolution of RTMT is shown on Figure 10 of the revised manuscript in addition to the evolution of RTLT. As indicated previously, the sum of RTLT+RTMT corresponds to the total residence time in the troposphere between 0 and 10km. A discussion has been added in section 3.4.2 of the revised manuscript on the influence of the middle troposphere into the hydrated profiles.**

**Figure 7 shows the summer averaged map of RTLT (filled contours) and DCCO (contours) for backtrajectories calculated over 48h (row 1), 96h (row 2), 120h (row 3) and 168h (row 4) from profiles with a mean RH in the 10-13km altitude range higher than 50%. RTLT from 46h backtrajectories is mostly located in the vicinity of Réunion Island, and the Northeast of Madagascar. The 96-hour RTLT pattern is significantly different and spreads over the East of the North of Madagascar for 2015 and 2016, and also West of Madagascar in 2014. The pattern of 120-hour and 168-hour RTLT is roughly similar to the 96-hour RTLT, except that RTLT is more spread over the Northeastern and Western regions of Madagascar. It means that most of the humid air masses**

**reaching the 10-13km layer above Reunion island were embedded in convective clouds and left the lower troposphere within 96 hours. The spread in the RTLT product from 96 hours to 168 hours backward in time is the result of horizontal atmospheric transport in the lower troposphere. Therefore, we can estimate an average time of transport between Reunion island and the main convective sources to be 96 hours. We added those comments in section 3.2 of the revised manuscript.**

[Figure]

_**Figure 7 : Average RTLT (filled contours) and DCCO (contours) from profiles in summer with a mean RH > 50% in the 10-13km altitude layer, calculated with 48h (row 1), 96h (row 2), 120h (row 3) and 168h (row 4) backtrajectories.**_

• Page 8, lines 252-253: I agree that the signature in the RH profiles around 11km could be potentially from the cyclone. But I'm also sure that there is also a cloud layer at 11 km altitude. Because the RHi is around 100% (RH eq. 60%) and the decrease in the humidity below the layer is slower than above the layer which indicates the hydration effect due to sedimented ice crystals. Here, I would suggest a bit more discussion about that.

**We agree with this and have added a discussion about this point in section 3.4.1 of the revised manuscript.**

• Page 9, lines 299-305 in combination with Figure 10: There are many peaks in RH which do not correspond to any signature in both trajectory products (e.g. around 01/Dec 2015, End of Jan 2015, beginning of summer 2015/2016 and many more). It would be good, if there would be more discussion in the text to explain those peaks and possible reasons (see also my main comment above).

**See our answers to your main comment**

• Page 9: In Section 3.4.3 the geographical origin of convection is discussed with the help of the product of DCCO and RTLT. In text the discussion is linked to the respective cyclones. It would really be helpful to follow the discussion, if

the names or at least the letters of the occurring cyclones (A-F) are written in each subplot of Figure 11. For example for March 2015 G and H or the full names of the cyclones.

**Figure 11 has been modified in the revised manuscript to include the names of tropical cyclones.**

**Technical comments/suggestions:**

**We took all technical comments into account.**

• Page 4, line 123: period of "the" study

• Page 4, line 126: to our treatment "of" convective

• Page 5, lines 141-142: "Residence times are computed using the model gridded output domain (1°x1°grid cells) values combining the results of the 3-hourly runs to provide a daily estimate of the source regions for air particles." This sentence is difficult to understand. Can you please rephrase.

• Page 5, line 146 and 147: "low" should be "lower"?

• Page 8, line 269: Skip the word "was"

• Page 9, lines 288-290: Sentence needs to be rephrased " estimated by correspon-des ".

• Please check the capitalization of the word Figure in text (e.g. page 8, line 273). It should be consistent throughout the manuscript.

• Figure 8: It would be good, if you could include the location of Reunion island in the maps.

**Anonymous Referee #3:**

We appreciate the thoughtful and constructive comments from the reviewers. Their helpful suggestions and attention to detail have made this a substantially better paper, and we greatly appreciate the time they put into the manuscript.

Please see below our responses (in bold) to the individual detailed comments. Numerous figures are shown in our response to illustrate our points but some are not included in the revised manuscript.

We have addressed all the reviewers 'comments and modified the manuscript and figures accordingly.

**Minor comments:**

- The authors could put their results a bit more into context, in particular when describing the seasonality of humidity and ozone in the subtropical region. For instance, the dry values in JJA as compared to DJF are connected to the ITCZ movement. Similarly, for the ozone only biomass burning is considered to explain the seasonality. But is there a role of other mechanisms such as transport from the stratosphere or from middle latitudes? This seems to be the case for the high ozone – low water vapour layers, which are ubiquitous over Reunion Island in JJA.

**Using ozonesonde and LIDAR from Réunion Island from 1998 to 2006, Clain et al. (2008) showed that the influence of stratospheric-tropospheric exchange induced by the subtropical jet stream is maximum in austral winter (June to August) when the jet moves closer to the island. They established that the 4-10 km and 10-16 km altitude ranges can be directly influenced by biomass burning and stratosphere-troposphere exchange. The influence of stratospheric-tropospheric exchange is in agreement with high ozone-low water vapor layers, which are ubiquitous over Réunion Island in austral winter. This discussion has been added in section 2.1 of the manuscript.**

- Given the important role of the MJO for deep convection over the Indian Ocean, its possible influence on convective activity deserves more discussion: you could at least identify the phases in the period under study and try to establish a connection.

**We have added some discussion in the text about the MJO status during the 3 austral summer periods studied. To define the state of the MJO, we used the Real-time Multivariate MJO (RMM) indices RMM1 and RMM2 data from the Australian Bureau of Meteorology (http://www.bom.gov.au/climate/mjo/graphics/rmm.74toRealtime.txt).**

[revised manuscript text omitted]

**We appreciate the thoughtful and constructive comments from the reviewers. Their helpful suggestions and attention to detail have made this a substantially better paper, and we greatly appreciate the time they put into the manuscript.**
**Please see below our responses (in bold) to the individual detailed comments. Numerous figures are shown in our response to illustrate our points but some are not included in the revised manuscript.**
**We have addressed all the reviewers 'comments and modified the manuscript and figures accordingly.**

**Major Comments:**
1. The RH data used comes from daily radiosonde measurements. The standard instruments used in radiosondes have long been known to suffer from significant dry biases in the upper troposphere and stratosphere due to instrument limitations and icing in supercooled liquid clouds (e.g., Miloshevich et al. 2004). No mention is given in the article on the quality of the RH measurements in the radiosonde data used and whether or not a correction has been applied to these data to account for known sources of dry bias. This is an important issue because it impacts much of the analysis presented.

**We agree that standard instruments used in radiosondes have long been known to suffer from significant dry biases in the upper troposphere and stratosphere. We compared the M10 measurements of RH (with respect to water) with Cryogenic Frospoint Hygrometer (CFH) water vapor sondes at the Maïdo Observatory (21.08°S, 55.38°E on the west coast of the island, 20 km away from the airport). Balloon-borne measurements of water vapor and temperature started in 2014 at the Maïdo Observatory on a campaign basis within the framework of the *Global* Climate Observing System (GCOS) *Reference* Upper-Air Network (*GRUAN*) network (Bodeker et al., 2015). The balloon sonde payload consists of the CFH and the Intermet iMet-1-RSB radiosonde for data transmission. The iMet-1-RSB radiosonde provides measurements of pressure, temperature, Relative Humidity (RH) and wind data (speed and direction from which zonal and meridional winds are derived). The CFH was developed to provide highly accurate water vapor measurements in the Tropical Tropopause Layer (TTL) and stratosphere where the water vapor mixing ratios are extremely low (~2 ppmv). CFH mixing ratio measurement uncertainty ranges from 5% in the tropical lower troposphere to less than 10% in the stratosphere (Vömel et al., 2007b); a recent study shows that the uncertainty in the stratosphere can be as low as 2-3% (Vömel et al., 2016). The CFH instrument is often launched in tandem with the Modem M10 sonde. The CFH RH data were calculated with the CFH water vapor mixing ratio and the Intermet iMet-1-RSB temperature using the water vapor pressure equation by Hyland and Wexler (1983) and interpolated to the same 200-m vertical grid as the M10 data. A total of 17 multiple-payload (CFH+M10) soundings is used for the comparison shown on Figure 1. The RH profiles from the CFH and Modem M10 show good agreement with differences of less than 10% mostly from the surface to the stratosphere. In the lower troposphere, below 5km, the mean RH difference is -1%. In the middle (5-10km) and upper troposphere (10-15km) the mean RH differences are 1.5 and 2.2% respectively. Near 15km, the M10 RH shows dry biases with a peak difference of -3.7% at 15.6km (the mean RH at this level ranges from 18.5 for the M10 sonde to 22.2% for the CFH sonde). Figure 1 is not shown in the revised manuscript but we have included some text on the comparison between the M10 and CFH sondes in section 2.1.**

[Figure]

***Figure 1: Vertical profiles of mean RH obtained from 17 multiple-payload sounding of CFH&M10 radiosondes (blue and black curves respectively on the left panel) at the Maïdo Observatory over 5 years (2014-2019). The mean profile of differences in RH is shown on the right panel.***

2. The use of a trajectory model to track air mass history and identify boundary layer sources of air associated with convection is a good approach to analysis, but the accuracy of the parameterized subgrid-scale motions along the trajectories is not well demonstrated. How well does the parameterized convection match actual convection in this region? The reliability of this approach is fundamental to the analysis and arguments presented in the paper and an assessment should be provided. The lack of agreement in RTLT and DCCO for the case study included in the paper (i.e., Figure 8) is particularly concerning as it suggests the parameterized convection fails to represent much of that observed (at least for the week shown). The only element of the paper acknowledging this potential issue (lines 299-305) is, in my opinion, insufficient.

[Figure]

*Figure 2: Correlations between mean upper-tropospheric water vapor mixing ratio (WV), residence times in the troposphere (0-5km RTLT, 5-10km RTMT and 0-10km RTLT+RTMT) and tropospheric residence times multiplied by DCCO. Backtrajectories over one-week were computed with the FLEXPART Lagrangian model.*

The location, intensity and vertical extent of deep convection in the FLEXPART model is determined by the calculation of a CAPE and the atmospheric thermodynamic profile using the meteorological fields from ECMWF. The trajectories are then redistributed vertically by a displacement matrix. Hence, the accuracy of the convective cells' location will be driven by the convective cells' locations within the ECMWF model output. An analysis on the correlations between WV mixing ratio and residence time calculated with 168h FLEXPART backtrajectories found at different altitude range in the troposphere is shown on figure 2. The first row shows the correlation between WV and RTLT (0-5km, left) or the product of RTLTxDCCO (right). According to the correlation calculations, 23% to 27% of the WV variability is explained byRTLT or RTLTxDCCO. Different phenomena can influence the upper tropospheric water vapor variabilty, such as deep convection, gravity waves or large-scale uplift of air masses. The low value of the R² factor is therefore difficult to interpret.

The second row on Figure 2 shows the correlation between WV mixing ratio and the rate of residence time in the middle troposphere (RTMT) between 5-10km altitude. The R-squared coefficient between RTMT and WV is 41.46%. The study of Schumacher et al. (2015) has shown that stratiform clouds have a vertical speed up to 10m s$^{-1}$ below 7km and then a slow ascent (<0.5 m s$^{-1}$) up to 10km. It suggests that a higher correlation between WV and in RTMT than in RTLT can be expected for these kinds of clouds.

**To conclude, the RTLTxDCCO product can represent only the influence of deep convective clouds, while higher R-squared coefficient with RTMT with WV shows that stratiform clouds can contribute to enhance the WV mixing ratio in the 10-13km range. We added comments in section 2.3 and 3.4.2.**

[Figure]

*Figure 3: Panel of RTLT (rows 1,3) and DCCO (rows 2,4) calculated using different duration of backtrajectories, initialized on 31 March 2014, during the passage of Tropical cyclone Hellen cyclone. Upper panel: backtrajectories over 48h (left) and 96h (right). Bottom panel: backtrajectories over 120h (left) and 168h (right).*

**Figure 3 presents the residence time in the lower troposphere for the case study of the Tropical cyclone Hellen calculated with 48h, 96h, 120h, and 168h FLEXPART backtrajectories. After 48h, no contribution in RTLT is found. After 96h, the RTLT is located north of the storm track, with an anti-clockwise dispersion toward Africa, outside the convective cells. It represents the fraction of air masses in the lower troposphere that was advected toward the convective clouds before reaching the 10-13km altitude range. The dispersion outside the convective cells increases with the 120h and 168h backtrajectories. Hence, it is clear that the colocation of RTLT with DCCO depends on the duration of the backtrajectories, and a poor value on RTLTxDCCO does not necessary mean that the trajectories were not lifted by deep convective clouds. The section 3.4.2 has been revised.**

3. There is a missed opportunity to put the case study of tropical cyclones into broader context. Previous work on the impact of tropical cyclones on upper troposphere and lower stratosphere water vapor and ozone is not acknowledged and would help in the authors' interpretation and argumentation here (e.g., Ray & Rosenlof, 2007; Zhan &Wang, 2012). Moreover, I do not find the impact of tropical cyclones on upper troposphere RH to be convincing in the paper, likely related to my concerns outlined in #2 above.

**We would like to point out that the study of Zhan & Wang in 2012 analyzed water vapor at a higher altitude range (Tropical Tropopause Layer between 14km and 20km) than our study.**
**However, the study of Ray and Rosenlof (2007) is relevant to our study and we thank you for pointing out this paper. A description of this study was added to the manuscript. Using AIRS and MLS satellite data, Ray and Rosenlof (2007) estimated the enhancement of water vapor within a 500 km radius of the center of 32 typhoons (Western Pacific) and 9 hurricanes (Northern Atlantic) at 223 hPa (~11km). They found an enhancement of up to 60 to 70 ppmv, within a 500 km radius north of the tropical storm centers, where the highest water vapor enhancement was found.**
**In the revised manuscript, we have included a list of tropical cyclones that have enhanced the WV in the upper-troposphere above Réunion Island. In 2014 summers the RTLT is influenced by tropical cyclones Hellen and Guito in the Mozambique Channel, Bejisa and Deliwe in the North East of Madagascar. In 2015, Bansi and to a lower degree Fundi in the Mozambique Channel, Chedza in the Northeastern part of Madagascar and in the eastern part of the basin, Haliba in the vinicity of Réunion Island have influenced the RTLT calculation.**
**The convective outflow of tropical cyclones that impacted the upper troposphere over Réunion Island was located south of the cyclone centers, the most hydrated part of the tropical cyclones in the Southern Hemisphere according to Ray and Rosenlof (2007).**
**Figure 4 presents the RTLT and DCCO for WV profiles that have been influenced by tropical cyclones in the South-West of the Indian Ocean in the previous 48h (row 1), 96h (row 2), 120h (row 3) and 168h (row 4). It is clear that the RTLT calculated with 48 and 96h backtrajectoires is collocated with DCCO of tropical cyclones. For longer backtrajectories (120h, 168h), RTLT covers a larger area in the lower troposphere, due to the advection of air masses in the lower troposphere.**

[Figure]

*__Figure 4: RTLT (filled contours) and DCCO (contours) calculated in the previous 48h (row 1), 96h (row 2), 120h (row 3) and 168h (row 4) for sounding associated with a tropical cyclone over the basin. Only the day with mean upper-tropospheric (10-13km) RH>50% have been selected.__*

**Specific Comments:**

Lines 134-135: "lower upper-tropospheric ozone values observed" is a bit confusing phrasing. Suggest rephrasing to "lower observed ozone values in the upper troposphere"

**Added in the revised manuscript.**

Line 144: suggest revising "the day" and "the latitude" to "day" and "latitude"Line 183: "affected by several three tropical cyclone events" Which is it several or three?

**We mean "three", it has been corrected**

Lines 194-195: Why is a threshold of 50% chosen? What is the sensitivity to this choice?

**A threshold on RH had to be chosen to isolate the ozone profiles that most likely were impacted by convection. The average WV mixing ratio between 10 and 13 km in austral summer (182ppmv) is larger than in austral winter (65ppmv), certainly due to the effect of deep convection and associated moister transport/cloudiness. The average RH of air masses with a WV higher than 182ppmv is 48.8%. Hence, we decided to use a threshold of 50% to isolate the profiles with anomalously high WV mixing ratio. The sensitivity to the value of this threshold is rather limited. Figure 5 shows the sensitivity of the ozone distribution to the RH threshold. The ozone distribution is very similar for RH thresholds ranging between 40% to 55%.**

[Figure]

*__Figure 5: Upper tropospheric ozone distributions for RH>40% (green), RH>45% (grey), RH>50%(blue), RH>55% (red), RH>60% (orange).__*

Lines 219-221: This statement doesn't seem appropriate. What if the responsible convection is land-based? One should expect a higher ozone mixing ratio in the boundary layer in that case. It seems reasonable that many/most convective sources for air in the upper troposphere at Reunion island would be land-based (e.g., look at Figure 3!).

**Thank you for the comment, we have modified our statement as follow:**
**"Another explanation would be that land-based convection (from Madagascar or the African continent) lifted air masses enriched in ozone from the boundary layer." The comment has been added in section 3.2.**

Line 221: comma should be a period
**Corrected**

Lines 343-346: as presented, this seems anecdotal and based on a single case. It Would be more convincing to show a map of the FLEXPART convective sources (i.e.,locations of most recent position in the lower troposphere) for matches with the RH profiles. It would help to better answer the question of importance of differences in boundary layer sources and mixing to impacting the upper troposphere ozone observed.

**Figure 4 of my previous comment and our answer to the major comment or referee #1 should address your comment.**

"Minnis et al. (2008) used the Moderate Resolution Imaging Spectroradiometer (MODIS) 11μm IR channel data and data taken by the Cloud-Aerosol Lidar and Infrared Pathfinder Satellite Observations (CALIPSO) to investigate the difference between cloud-top altitude Ztop and infrared effective radiating height Zeff for optically thick ice cloud (i.e. deep convective clouds). They found an error of 2km in the derived cloud top altitude from passive sensors for clouds higher than 14km in altitude, and an error of 1.25km below. This suggests that using a threshold of 230K to define deep convective clouds can induce an error in the selection of these clouds. Thin cirrus clouds could be included in our

selection of deep clouds but it is difficult to say how much by using passive satellite sensors only. Additional measurements from active sensors such as CALIPSO would be required to distinguish between the deep convective cores inferred from passive infrared radiances and cold in situ formed cirrus clouds. However, this is beyond the scope of this study. We tested the sensitivity to the 230K threshold, and found that our definition of DCCO with a brightness temperature of 230K is a good compromise to distinguish deep convective clouds over land and ocean."

**A clearer description of the ECMWF fields used for the initialization of FLEXPART is describe as follow:**

**Lines 155-159 page 5:**

"We use input meteorological fields from the ECMWF Integrated Forecast System (IFS, current ECMWF operational data) that have 137 vertical levels up to 0.01hPa. The vertical resolution varies from ~20 m near the surface, ~100 m in the low troposphere, ~300 m in the middle/upper troposphere and 500 m in the stratosphere. FLEXPART was driven by using operational ECMWF analysis at 00, 06, 12 & 18 UTC and the 3 & 9-hour forecast fields from the 00 and 12 UTC model analysis."

**We added a clearer description of the FLEXPART output as follow (asked by AR#1):**

**Lines 165-170, page 5:**

"FLEXPART backtrajectories can then be processed in the form of a gridded output of the residence time. The residence time field was reported on a regular 0.5°x0.5° output grid every 3 hours. The resolution of the gridded output is independent from those of the meteorological input. Therefore, we use 0.25°x0.25° operational ECMWF input fields to compute the backward trajectories and the resulting residence time is reported on a regular 0.5°x0.5° output grid."

**We provide more details on the convective parametrization used in FLEXPART as suggest by AR#2:**

**Lines 179-184, pages 5-6:**

"The location, intensity and vertical extent of deep convection in the FLEXPART model is determined by the calculation of a CAPE and the atmospheric thermodynamic profile using the meteorological fields from ECMWF. The trajectories are then redistributed vertically by a displacement matrix. Hence, the accuracy of the convective cells' location will be driven by the convective cells' locations within the ECMWF model output."

**In section 3.1, we discuss the RHi and water vapor mixing ratio measured in the upper-troposphere by the Meteomodem M10 sonde. The discussion provides details on air masses with a RHi of 100% and water vapor mixing ratio associated with RH>50% profiles. In Fig. 3 we added the 0°C isotherm in red and the RHi>100% in black contours to identify air masses saturated with respect to ice. Fig. 4 has been edited with a color scale to link the RH evolution with the RHi (top) and the water vapor mixing ratio (Bottom). These changes were suggested by AR#1.**

**Lines 201-223, pages 6-7:**

[revised manuscript text omitted]

**We added the RHi profile in Fig. 8c and a discussion on the measured RHi for the case study of tropical cyclone Hellen (in response to AR#1 comment).**

**Lines 346-348, page 10:**

"Since RHi is around 100% at 11 km and the decrease in humidity below the layer is slower than above the layer, it probably indicates a hydration effect due to sedimented ice crystals."

**From AR#1 and AR#2 comments, we decided to further discuss the RTLT distribution for the case study of tropical cyclone Hellen.**

**Lines 359-366, page 10:**

"A detailed analysis of RTLT was performed for tropical cyclone Hellen with different residence time in the lower troposphere with 48 hours, 96 hours, 120 hours, and 168 hours FLEXPART backtrajectories (not shown). After 48 hours, no contribution in RTLT is found. After 96 hours, the RTLT is located north of the storm track, within the convective region of tropical cyclone Hellen. After 120 and 168 hours, an anti-clockwise dispersion toward Africa, outside the convective cells is found. It represents the fraction of air masses in the lower troposphere that was advected toward the convective clouds before reaching the 10-13km altitude range. Hence, the collocation of RTLT with DCCO depends on the collocation of the convective regions in FLEXPART+ECMWF and METEOSAT7, but also on the duration of the backtrajectories."

**The effect of tropical cyclones on upper tropospheric water vapor transport over the Southwest Indian Ocean is compared with the study of Ray and Rosenlof (2007). The comparison was suggested by AR#2 to put the study of tropical cyclones into a broader context.**

**Lines 401-407, page 11:**

"The results are consistent with the study of Ray and Rosenlof (2007). Using Atmospheric Infrared Sounder (AIRS) and MLS satellite data, Ray and Rosenlof (2007) estimated the enhancement of water vapor due to 32 typhoons (Western Pacific) and 9 hurricanes (Northern Atlantic) at 223 hPa (~11km). They found an enhancement of up to 60 to 70 ppmv, within a 500 km radius north of the tropical storm centers, where the highest water vapor enhancement was found. The convective outflow of tropical cyclones that impacted the upper troposphere over Réunion Island was located south of the cyclone centers, the most hydrated part of the tropical cyclones in the Southern Hemisphere according to Ray and Rosenlof (2007)."

**We added the sum of fraction of residence time in the middle troposphere (sRTMT) in figure 10, which induces major changes in part 3.4.2. The study of sRTMT improves significantly the analysis and gives more details about the air masses origin (discussion in our responses to AR#1 and AR#2).**

**Lines 408-419, page 12:**

[revised manuscript text omitted]

 On the two panels, the black thick line corresponding to the monthly mean RH in the upper-troposphere is added.

**Figure 5:**

We have added the mean RH for hydrated profile (WV>121ppmv) with a dotted red line and for strong hydration (WV>2.5x121ppmv) with a red line.

**Figure 6:**

On the left panel we added the position of Réunion Island (white star) and center of tropical cyclone Hellen on 31 March 2014 (red star).

RHi profiles are added in the right panel (the daily profile and average over the austral summers 2013 to 2016). The isotherm 0°C is also added.

**Figure 10:**

We have added the sRTMT variation in the three figures. The RH variation, which could depend on temperature is replaced by the water vapor mixing ratio in purple.

**Figure 11:**

[revised manuscript text omitted]